# Optogenetic therapy: high spatiotemporal resolution and pattern discrimination compatible with vision restoration in non-human primates

Gregory Gauvain [1✉], Himanshu Akolkar[1,2], Antoine Chaffiol[1], Fabrice Arcizet[1], Mina A. Khoei[1], Mélissa Desrosiers[1], Céline Jaillard[1], Romain Caplette[1], Olivier Marre [1], Stéphane Bertin[3], Claire-Maelle Fovet [4], Joanna Demilly[4], Valérie Forster[1], Elena Brazhnikova[1], Philippe Hantraye[4], Pierre Pouget[5], Anne Douar[6], Didier Pruneau[6], Joël Chavas[6], José-Alain Sahel [1,2,3], Deniz Dalkara[1], Jens Duebel [1], Ryad Benosman[1,2] & Serge Picaud [1✉]

Vision restoration is an ideal medical application for optogenetics, because the eye provides direct optical access to the retina for stimulation. Optogenetic therapy could be used for diseases involving photoreceptor degeneration, such as retinitis pigmentosa or age-related macular degeneration. We describe here the selection, in non-human primates, of a specific optogenetic construct currently tested in a clinical trial. We used the microbial opsin ChrimsonR, and showed that the AAV2.7m8 vector had a higher transfection efficiency than AAV2 in retinal ganglion cells (RGCs) and that ChrimsonR fused to tdTomato (ChR-tdT) was expressed more efficiently than ChrimsonR. Light at 600 nm activated RGCs transfected with AAV2.7m8 ChR-tdT, from an irradiance of $10^{15}$ photons.cm$^{-2}$.s$^{-1}$. Vector doses of $5 \times 10^{10}$ and $5 \times 10^{11}$ vg/eye transfected up to 7000 RGCs/mm$^2$ in the perifovea, with no significant immune reaction. We recorded RGC responses from a stimulus duration of 1 ms upwards. When using the recorded activity to decode stimulus information, we obtained an estimated visual acuity of 20/249, above the level of legal blindness (20/400). These results lay the groundwork for the ongoing clinical trial with the AAV2.7m8 - ChR-tdT vector for vision restoration in patients with retinitis pigmentosa.

---

[1] Sorbonne Université, INSERM, CNRS, Institut de la Vision, 17 rue Moreau, F-75012 Paris, France. [2] Department of Ophthalmology, University Pittsburgh Medical Center, Pittsburgh, PA, USA. [3] CHNO des Quinze-Vingts, INSERM-DGOS CIC 1423, 28 rue de Charenton, F-75012 Paris, France. [4] Département des Sciences du Vivant (DSV), MIRcen, Institut d'imagerie Biomédicale (I2BM), Commissariat à l'Energie Atomique et aux Energies Alternatives (CEA), 92260 Fontenay-aux–Roses, France. [5] ICM, UMRS 1127 UPMC – U 1127 INSERM – UMR 7225 CNRS, Paris, France. [6] Gensight Biologics, 74 rue du faubourg Saint Antoine, F-75012 Paris, France. ✉email: gregory.gauvain@inserm.fr; serge.picaud@inserm.fr

Optogenetics has transformed neurobiology, by enabling scientists to control the activity of excitable cells with light[1]. Optogenetic therapy has also raised considerable hopes for new forms of brain–machine interfaces, with cell selectivity and distant optical control. Rebuilding vision through optogenetic approaches is conceptually straightforward, as the aim is to restore light sensitivity in the residual retinal tissue after photoreceptor degeneration, in diseases such as retinal dystrophies[2] and age-related macular degeneration[3]. These diseases mostly affect photoreceptors, so the remaining retinal layers, including the retinal ganglion cells (RGC) can still communicate with the brain via the optic nerve. The feasibility of reactivating these retinal layers has already been demonstrated with retinal prostheses[4,5] despite their major limitations in terms of surgery, spatial resolution, and cell specificity[6].

The use of optogenetics to restore vision was first proposed by Zhao Pan and his colleagues[7,8]. They expressed the microbial opsin channelrhodopsin-2 (Chr2) in the RGCs of blind mice[7], and subsequently in the retina of normal marmosets[8]. These studies led to a clinical trial using this microbial opsin, which began in February 2016, but for which no results have yet been published[9]. Other retinal cells (bipolar cells[10–12] and dormant cone photoreceptors[13,14]) were subsequently targeted to restore vision in blind rodents, postmortem retinal tissue, and non-human primates. Clinically, the choice of cell type targeted depends on the stage of tissue remodeling after photoreceptor degeneration[15–17]. We performed translational studies targeting retinal ganglion cells (RGCs), the neurons projecting their axons out of the retina because this strategy could potentially work in all patients who have lost their photoreceptors, regardless of disease stage[18].

RGCs in the non-human primate retina can be activated with a more sensitive form of Chr2, "CatCH"[19]. We tested this approach with an RGC-specific promoter[20]. However, the intensity of blue light required was close to radiation safety limits[21]. It was therefore of clinical importance to evaluate other opsins potentially conferring a better balance between light sensitivity and channel kinetics[22].

In this study, we used the optimum AAV capsid with the most red-shifted opsin, operating at a wavelength 45 nm longer than ReaChR[23]. We demonstrate here that the high spatiotemporal resolution of this system is suitable for use in vision restoration. A single intravitreal injection, at a dose of $5 \times 10^{10}$ or $5 \times 10^{11}$ vg/eye transfects up to 7000 RGCs/mm$^2$ in the perifovea. Responses were elicited at a stimulus duration of 1 ms and saturated at a stimulus duration of 30–50 ms. Furthermore, using the responses to moving bars and letters generated on a multielectrode array, we obtained an estimated theoretical visual acuity of 20/249, which is above the threshold for legal blindness. These characterizations of the visual response in the non-human primate retina paved the way for the ongoing clinical trial with the AAV2.7m8-ChrimsonR-tdT vector for vision restoration in patients with retinitis pigmentosa.

## Results

**AAV2.7m8–ChR-tdT provides the highest transduction efficiency.** Our primary objective was to determine the best genetic construct for expressing ChrimsonR in primate RGCs. The intravitreal delivery of AAV vector in non-human primates (NHPs) has been shown to lead to transduction of the ganglion cell layer in the perifoveal ring[20,24]. The mutated capsid AAV2.7m8 has been demonstrated to yield stronger transduction of the perifovea[25]. We, therefore, decided to compare the efficiency of ChrimsonR (ChR) expression from the AAV2.7m8 vector with that of the wild-type AAV2. ChR is often fused to the

fluorescent protein tdTomato for visualization of its expression within cells. We therefore also investigated whether the native ChR protein and the ChrimsonR-tdTomato (ChR-tdT) fusion protein were produced in similar amounts in primate RGCs. The four selected constructs (AAV2 and AAV2.7m8 vectors encoding either ChR or ChR-tdT) were each injected into four eyes, at the same concentration ($5 \times 10^{11}$ vg/eye); eight animals in total were used for this experiment (Supplementary Table S1). While no in-depth behavior analysis was performed, none of the treated animals displayed signs of photophobia or vision-related changes in behavior under normal lighting in the animal house. The level of microbial opsin expression was assessed in functional analyses two months after the intravitreal injection in vivo. The transduced retinas were isolated ex vivo and divided into hemifovea for extracellular large-scale 256-multielectrode array (256-MEA) recordings for one hemifovea and two-photon targeted patch-clamp recordings for the other hemifovea (Fig. 1). No natural light responses were recorded in our experimental conditions, but we nevertheless added synaptic blockers to the bath to suppress any residual natural light responses (see Supplementary Materials and Methods). For the quantitative measurement of functional efficacy, the results shown are the multiunit activity on all electrodes following full-field stimuli; the use of this approach may have amplified the differences between results (see below). 256-MEA recordings revealed large differences in the ability to generate functional ChR expression between vectors (Fig. 1a–d). Recording quality was defined as the number of electrodes for which spontaneous spiking activity could be measured (active electrodes: $152 \pm 46$ electrodes per retina explant, on average), whereas ChR efficacy was defined as the number of electrodes displaying an increase in activity during the presentation of light flashes (responsive electrodes, SN ratio >4). This quantification revealed the existence of a significant difference between the constructs, with the highest efficacy for AAV2.7m8–ChR-tdT (Fig. 1c, 64.4% of active sites responsive vs. 13.4%, 10.6 and 0% for AAV2.7m8–ChR-tdT, AAV2.7m8–ChR, AAV2–ChR-tdT, and AAV2–ChR, respectively, $P < 0.001$). For all constructs, the foveal area was identified and selected for recording. The corresponding retinal explant was positioned on the MEA before confirmation of the eventual presence of fluorescence. If no light response was measured, we repositioned the tissue on the array to increase the sampling area. Light sensitivity was measured with a range of light intensities, from $1.37 \times 10^{14}$ to $6.78 \times 10^{16}$ photons cm$^{-2}$ s$^{-1}$ on all responsive retinas (Fig. 1b, d). Responses were obtained with AAV2.7m8–ChR-tdT, in all four retinas tested with this construct (verses 2, 1, and 0 for AAV2.7m8–ChR, AAV2–ChR-tdT, and AAV2–ChR, respectively). This vector also yielded the highest light sensitivity, with responses recorded for $2.34 \times 10^{15}$ photons cm$^{-2}$ s$^{-1}$, at a frequency higher than for the other constructs.

Consistent with its optogenetic origin, the spiking activity had a short latency, was activated for the whole duration of stimulation and its frequency was modulated by light irradiance. Furthermore, an increase in the number of responsive electrodes with increasing irradiance was clearly observed (Fig. 1b). We recorded the action spectrum of the responses (Fig. 1e), and the measured peak was consistent with the known spectral sensitivity of ChR, at about 575 nm[23].

The results of 256-MEA experiments were confirmed in two-photon targeted patch-clamp recordings (Fig. 1f–h) on the other hemifovea from the same eye. At the highest irradiance, AAV2.7m8–ChR-tdT elicited robust responses, with a typical photocurrent shape, consisting of a fast transient followed by a steady-state current (Fig. 1g, 12 to 375 pA, mean: $88.7 \pm 25.5$ pA, $n = 17$). These currents increased steadily with increasing light intensity, from $5.8 \times 10^{14}$ to $3.15 \times 10^{17}$ photons cm$^{-2}$ s$^{-1}$ (Fig. 1h).

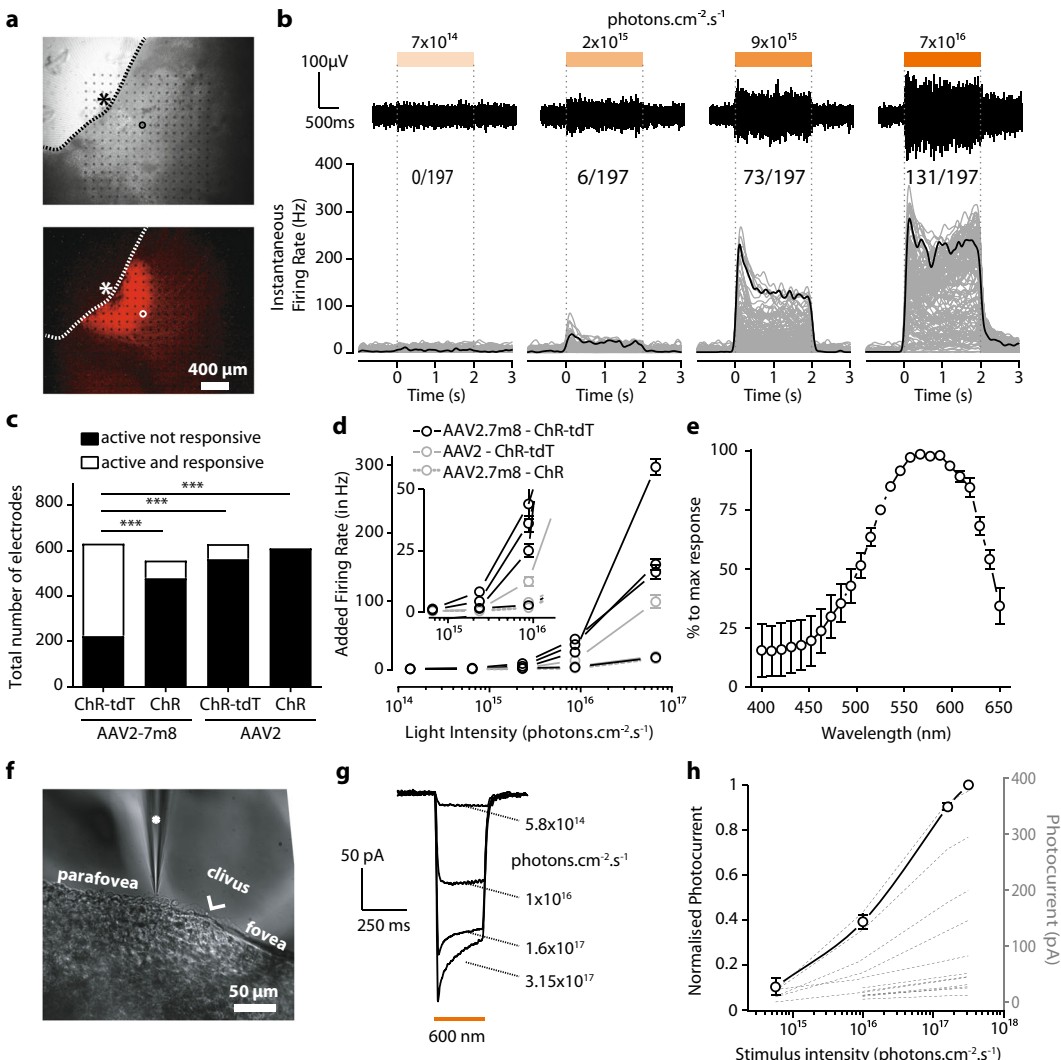

**Fig. 1 Higher transfection efficiency with AAV2.7m8–ChR-tdT in NHP retinas. a** Images of a primate retina expressing AAV2.7m8–ChR-tdT as observed during MEA recordings. Top: Infrared image, electrodes can be seen as black dots; the retina explant in gray, and its limit is shown as a dashed line. An asterisk indicates the center of the fovea, the circle indicates the example electrode in **b**. Bottom: Epifluorescence image of the same piece of the retina. The strong perifoveal expression can be observed in the mounted hemifovea thanks to tdT fluorescence. **b**, top: Raw signal recorded from one sample electrode (circled electrode in **a**) in response to stimuli of increasing intensities ($7 \times 10^{14}$, $2 \times 10^{15}$, $9 \times 10^{14}$, and $7 \times 10^{16}$ photons cm$^{-2}$ s$^{-1}$). Light to dark orange rectangles indicate the temporal duration (2 s) of the different intensities of light stimulation delimited by dashed vertical lines. Voltage and temporal scale on the left (bottom) Spike density function for all the active electrodes of the hemifovea shown in a (gray lines, $n = 197$ lines) as a function of time, before, during, and after a two-second stimulus. Firing rates were averaged over ten repetitions. Black lines show the mean firing rate for the electrode displayed in the upper panels and circled in **a**. The numbers at the top indicate the number of responsive electrodes per light intensity compared to total active electrodes (i.e., electrodes where spikes are recorded). **c** Total of active electrodes recorded for the four different constructs (four experiments per construct, theoretical maximum: 1024 electrodes per construct). Data are then split for each construct between active and responsive electrodes (white) and active but unresponsive electrodes (black). The proportion of active and responsive electrodes is maximal for AAV2.7m8–ChR-tdT ($n = 4$, Fisher contingency test, $P < 0.0001$). **d** Mean additional firing rate per responsive retina for the four constructs ± SEM (four responsive retinae for AAV2.7m8–ChR-tdT, two for AAV2–ChR-tdT, one for AAV2.7m8–ChR, zero for AAV2–ChR). Stimulation at 590 nm ±15 nm. The inset shows a zoomed image around the first responsive intensity: $2.34 \times 10^{15}$ photons cm$^{-2}$ s$^{-1}$. **e** Mean normalized action spectrum for three retinas expressing AAV2.7m8–ChR-tdT ± SEM. **f** Infrared image of the perifoveal region from a retina treated with AAV2.7m8–ChR-tdT and recorded by two-photon targeted patch clamp. The patch-clamp electrode is indicated with a white asterisk, the clivus ocularis is indicated, separating the fovea from the parafovea. **g**, **h** Whole-cell patch-clamp recordings of ChR-tdT-expressing macaque perifovea neurons. **g** Photocurrent traces from one recorded cell at different light intensities. **h** ChR-induced photocurrents peaks are represented as a function of light intensity for each individual recorded cell (dashed lines, $n = 17$), the solid line represents the population-averaged photocurrent after normalization to maximal peak value +/− SEM. Light stimulation intensity ranged from $5.8 \times 10^{14}$ to $3.2 \times 10^{17}$ photons cm$^{-2}$ s$^{-1}$.

With the AAV2.7m8–ChR-tdT combination, we recorded 18 responsive cells (5, 0, 7, 6 cells/retina), whereas only four responsive cells (0, 3, 1, 0 cells/retina) were obtained with the AAV2–ChR-tdT construct. In the absence of tdT fluorescence, for AAV2.7m8–ChR and AAV2–ChR, extracellular recordings were performed on

random healthy RGCs in the perifoveal area (>40 cells per condition). In these conditions, none of the RGCs for which recordings were made displayed light-evoked responses, even under conditions known to activate ChR. We cannot exclude a potential bias in favor of the construct including tdTomato, particularly in

the patch-clamp experiments, but the positioning of the MEA based on foveal identification probably rule out such a bias in MEA recordings. These MEA recordings were consistent with greater efficacy of the AAV2.7m8–ChR-tdT constructs; this construct was therefore used in all subsequent experiments.

**AAV2.7m8–ChR-tdT provides greater light sensitivity at a dose of $5 \times 10^{11}$ vg/eye.** Once the capsid and genetic payload had been selected, we assessed transgene stability over time (6 months). In the same set of experiments, we optimized virus load, using three different amounts of vector for intravitreal delivery: $5 \times 10^9$, $5 \times 10^{10}$, and $5 \times 10^{11}$ vector genomes per eye (vg/eye), for a total of six animals (four eyes per dose). Additional results were obtained with four more retinas treated with the high dose. After the injections, we examined the eyes of the animals monthly for posterior uveitis and vitreal haze. Clinical evaluation showed no significant immune response following ChR-tdT expression (Supplementary Fig. S1). The success of our vision restoration strategy depends on: (1) a large, dense area of transfected cells and 2) high light sensitivity per cell. For correct estimation of the number of cells transfected and of the retinal coverage of expression, we performed manual cell counts on RGC layers in the confocal stack of images for hemifoveas. We used these counts to establish density maps (Fig. 2a) and density profiles (Fig. 2b). The number of ChR-tdT-expressing cells increased with increasing vector dose (mean total number of transfected cells: $491 \pm 64$, $4395 \pm 631$, and $5935 \pm 715$, for ChR-tdT at $5 \times 10^9$, $5 \times 10^{10}$, and $5 \times 10^{11}$ vg/eye, respectively, see "Methods"). The local densities achieved with the two highest concentrations were not significantly different (Fig. 2a), but eyes receiving $5 \times 10^{11}$ vg expressed ChR-tdT with a moderately higher eccentricity (Fig. 2b), resulting in expression over a potentially larger area for this dose. Based on the automatic counting of DAPI-stained nuclei in the same samples, we estimated the peak density at ~40,000 RGCs/mm$^2$, with an eccentricity of 0.4 mm (Supplementary Fig. S2), consistent with previous RGC density maps[26]. Based on this number, we estimated that ~20% of RGCs expressed ChR-tdT. Before fixation these hemifoveas were used for MEA recordings, to assess light sensitivity following long-term expression (Fig. 2c). In terms of the fraction of responsive electrodes, there was no clear difference between $5 \times 10^{10}$ and $5 \times 10^{11}$ vg/eye, but the number of retinas with responsive electrodes was smaller for the lower dose (only one of four retinas with responsive electrodes, Fig. 2d). More importantly, the different viral doses led to different levels of light sensitivity, with $5 \times 10^{11}$ vg/eye yielding the strongest overall responses and the lowest response threshold (Fig. 2e, see Supplementary Table S2 for Tukey's multiple-comparison test). We cannot exclude the possibility of a decrease in the number of ChR-tdT-expressing cells between 2 and 6 months, as we were unable to obtain cell counts for both time points. However, we observed no major differences in the expression profile on the fovea and no changes in the subcellular pattern of expression (Supplementary Fig. S3). Furthermore, we observed no difference in the fraction of responsive electrodes (Fig. 2d, $102 \pm 58$ vs. $73 \pm 65$ for 2 months and 6 months, respectively, for $5 \times 10^{11}$ vg/eye), or light sensitivity (Figs. 1d and 2e). Based on these findings and the absence of a significant immune response to viral load or ectopic gene expression (Supplementary Fig. S1), we selected $5 \times 10^{11}$ vg/eye as the most appropriate dose for our therapy. Thus, all the data presented hereafter are for a dose of $5 \times 10^{11}$ vg/eye after 6 months of expression.

**Activity modulation at the millisecond scale.** Natural vision is dependent on a highly dynamic temporal range of information for the perception of moving objects. In virtual reality goggles, the minimum information transfer mode seems to be dependent on the

video rate (30 Hz[27]). Vision restoration for locomotion or for the perception of dynamic scenes should therefore restore light sensitivity to at least this temporal scale. We, therefore, measured the temporal dynamics of our optogenetic responses with full-field monochromatic stimuli ($2 \times 10^{17}$ photons cm$^{-2}$ s$^{-1}$ at 600 nm $\pm$ 10 nm) of increasing duration (1–2000 ms). This light intensity was selected because it generated the highest firing rates while remaining below radiation safety limits for continuous eye exposure (~$6 \times 10^{17}$ photons cm$^{-2}$ s$^{-1}$[28,29]). Significant light responses were detected for stimulus durations as short as a few milliseconds (Fig. 3a). Interestingly the firing rate of RGCs reached a plateau for durations of 30–100 ms, depending on the retina tested (Fig. 3b). We defined the minimal stimulus duration generating a reliable response, by calculating the time to the first spike after the onset of stimulation, for all responsive electrodes (Fig. 3c). For stimuli lasting 5 ms or more, we observed a median time to a first spike of about 9 ms. Stimulation for 5 ms is, therefore, sufficient for the reliable activation of RGCs reliably, and the intracellular integration of the ChR-tdT photocurrent initiated spiking in less than 10 ms for most of the responsive electrodes. Furthermore, for a stimulation duration of 20 ms, the time to first spike was between 3 and 11 ms for 50% of the responsive electrodes. We then looked at the distribution of firing rates following stimulations of increasing duration (Fig. 3d). Even for 1 ms stimuli (Fig. 3c, d, dark-blue curves), 12% of electrodes measured a peak firing rate exceeding 100 Hz. We considered multiunit recording, but this observation indicates that, for some RGCs, a 1 ms stimulus was sufficient to elicit a strong response, as clearly seen in Fig. 3a. For stimuli lasting 5 and 20 ms, 48% and 69%, respectively, of the responsive electrodes had firing rates above 100 Hz. Finally, for the longest stimulus duration tested (2 s), peak responses and the sustained firing rate decreased during consecutive stimulations (Fig. 3a, b, d), but both these parameters subsequently recovered. We investigated this effect further for long stimulation durations, by calculating the Fano factor, a measurement of the variability of spike number relative to the mean number of spikes, for all electrodes, as a function of stimulation duration. The Fano factor was below 1 for the short duration (1–200 ms), indicating a lower variability than for the Poisson distribution, but we observed a large increase in spike train variability for stimulations lasting 2 s (Fig. 3e). Most of this effect can be attributed to stimulus hysteresis, as retinal sensitivity subsequently recovered. Consistent with this observation, recordings of activity in response to achromatic binary white noise with a 50% pseudorandom selection rate revealed a gradual decline of evoked activity. The underlying mechanism of this modulation may involve an inactivated state of the microbial opsin[30] or the inactivation of the voltage-gated channels in the ganglion cells. A simple monochrome transformation of natural images would result in a large number of pixels with high values (i.e., light gray) potentially leading to rapid deactivation of retinal ganglion cells. The goggles used for visual stimulation include an event-based asynchronous camera outlining object contours[31]. It should therefore be possible to overcome the problem of retinal ganglion cell deactivation by reducing the number of active pixels in a projected frame through the limitation of active pixels to object contours. Flickering stimuli should be used, to reduce the total amount of light and the risk of an increase in spike train variability. For a light pulse width between 5 and 20 ms, stimulation frequencies between 100 and 25 Hz could be used.

**ChR-tdT can produce a high temporal photocurrent and spiking modulations.** In parallel with our population study on MEA, we investigated temporal dynamics, at the single-cell level, by recording photocurrent modulation in single cells. In all recorded hemifovea, fluorescent transfected cells could be seen in the characteristic half-torus shape (Fig. 4a, b). Using

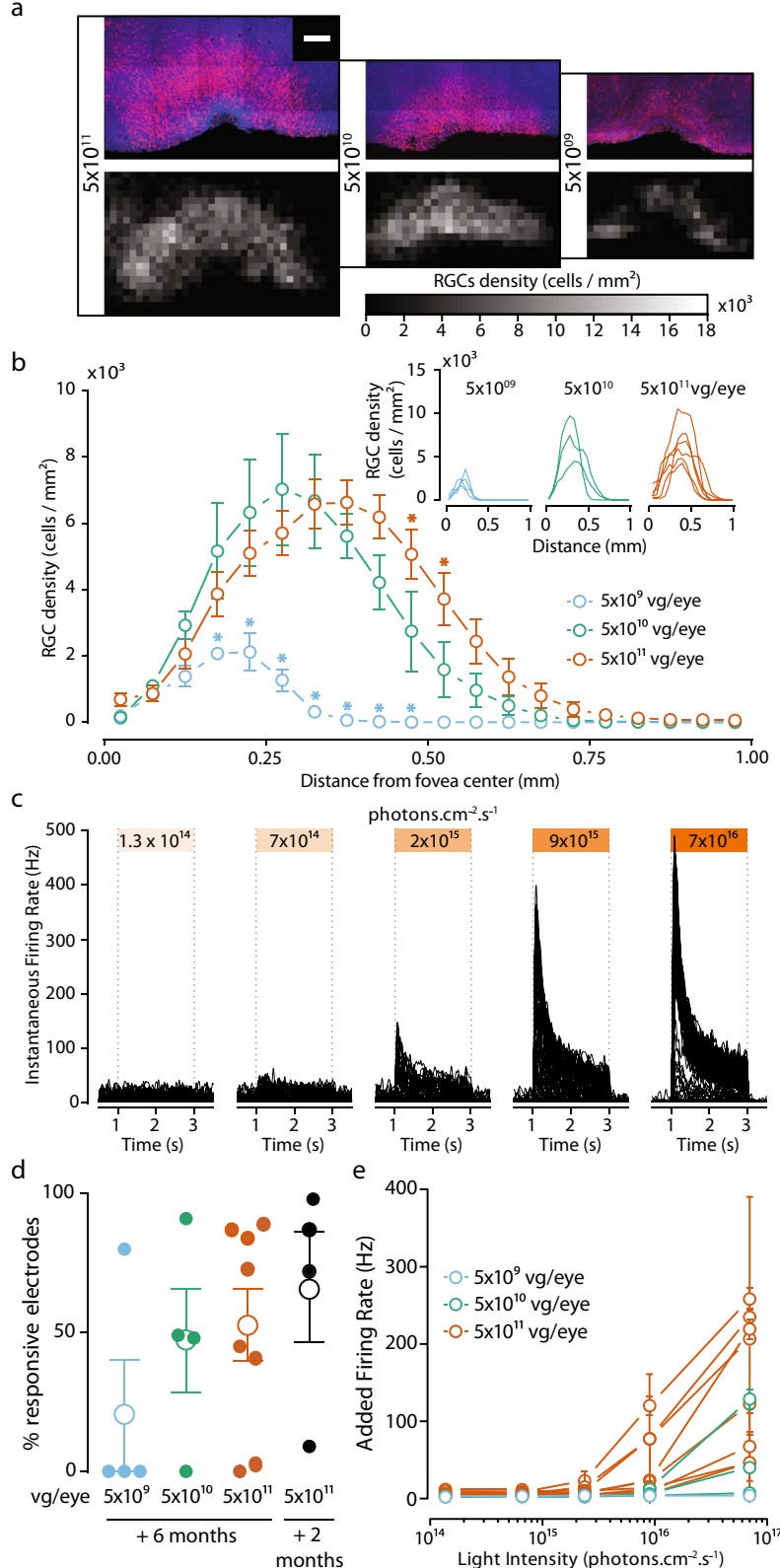

two-photon guided patch-clamp techniques, we obtained recordings for ChR-tdT-expressing RGCs in the perifoveal area with a cell-attached or voltage-clamp intracellular configuration (Fig. 4c). We first replicated the analysis of photocurrent modulation by light intensity, comparing responses at 6 months (Fig. 4c–e) and 2 months of expression (see Fig. 1f–h). The

mean normalized photocurrent followed a similar photo-sensitivity curve at two and six months (Fig. 4d), with an activation threshold in the $10^{15}$ photons $cm^{-2}\,s^{-1}$ intensity range, and robust responses to light stimuli at a wavelength of 600 nm (±10 nm) well below the illumination radiation safety limits for the human eye (~$6 \times 10^{17}$ photons $cm^{-2}\,s^{-1}$)[21,29,32].

**Fig. 2 AAV2.7m8–ChR-tdT induces high-density long-term expression in the perifoveal area in non-human primates. a**, top: Projections of confocal stack stitches showing perifoveal areas of retinas treated 6 months earlier with three different doses of the vector ($5 \times 10^{11}$, $5 \times 10^{10}$, and $5 \times 10^{9}$ vg/eye, respectively left to right). ChR-tdT-expressing cells are shown in red, whereas DAPI staining of the nuclei is shown in blue. Scale bar: 200 µm. Bottom: Density maps of ChR-tdT-positive RGCs for the three hemifoveas showed on top. **b** Density profiles of ChR-tdT-expressing RGCs relative to retinal eccentricity for the three vector doses tested. Density is expressed as the mean value for all retinas analyzed ($P < 0.05$, two-way ANOVA, multiple comparisons). The individual retina profile is shown in the inset ($n = 4$ for $5 \times 10^{9}$, $n = 3$ for $5 \times 10^{10}$, and $n = 6$ for $5 \times 10^{11}$ vg/eye). **c** Spike density function for all responsive electrodes of a retina, treated with $5 \times 10^{11}$ vg, in response to different light levels. **d** Responsive electrode fraction, measured in MEA experiments, for the three doses after 6 months of expression, compared to 2 months of expression for $5 \times 10^{11}$ vg.eye (same data as Fig. 1); no significant differences shown (Wilcoxon–Mann–Whitney test). Filled circle: value for an individual retina; open circle: mean for all the retinas ± SEM. **e** Mean±SEM additional firing rate per responsive retina for the three doses, at different light levels, at 590 nm ±15 nm, see Supplementary Table S2 for the statistical analysis of the dose-dependent response.

In comparisons of peak photocurrent or peak firing rate at maximal light intensity ($3.15 \times 10^{17}$ photons cm$^{-2}$ s$^{-1}$), we found no significant difference between the two durations of expression (Fig. 4e, 2 and 6 months), suggesting that ChR-tdT expression remained stable for as long as 6 months.

We then investigated response kinetics, by recording responses to light stimuli of increasing durations (Fig. 4f) in the cell-attached mode (spikes) and in the whole-cell configuration (photocurrent). The photocurrent and the spike rate both precisely followed stimulus components, such duration (Fig. 4f) and frequency (Fig. 4g, h) precisely. Interestingly, the decrease in photocurrent amplitudes, from the initial peak to the lower sustained amplitude, was paralleled by a similar decrease in firing rates. We further investigated the effect of flicker stimuli in a 50% duty cycle (half the stimulus period with the light ON, at 2–28 Hz) or at a specific stimulus duration (5 ms, from 10 to 100 Hz) (Fig. 4g–i). For full duty-cycle stimulation (Fig. 4g), the photocurrent closely followed the stimulus frequency, for flicker stimulations of up to 30 Hz. These results are consistent with the rapid opening and closing kinetics of the ChR channel in RGCs (10 to 90% rise time, 5.2 ± 1.7 ms; decay time, 27 ± 2.9 ms for a stimulus duration of 250 ms at $3.15 \times 10^{17}$ photons cm$^{-2}$ s$^{-1}$, $n = 5$, Fig. 4c). The fast photocurrents allow neurons to translate each light pulse robustly into a burst of spikes, but the decay time of the photocurrent does not allow a complete return to the resting level during trains of the stimulus (e.g., 30 Hz flicker, Fig. 4g). We then used a lower duty cycle, consisting of trains of 5 ms stimuli (20 pulses at frequencies between 10 and 100 Hz), which has been shown to activate ChR-tdT-expressing RGCs in MEA experiments (Fig. 3). With such short stimuli, our recordings showed that photocurrent could be modulated at high frequencies, with large amplitudes (50–100 pA), generating periodic spiking activities. The cells for which recordings were obtained followed the stimulus train precisely, even at 100 Hz, but current deactivation was incomplete between light pulses (Fig. 4h, i). Recordings in cell-attachment mode confirmed the ability of neurons to follow stimulus frequencies of up to 66 Hz, despite incomplete current deactivation (Fig. 4i). This 60 Hz range is compatible with the flicker perception limits observed for natural vision in human subjects[33,34] and could potentially be used for fast video rate stimulation in human patients. Finally, we activated cells with a stimulus simulating natural properties: a one-dimensional random walk, and consisting of a rapidly changing contrast stimulus (full-field stimulus with intensities ranging from $3 \times 10^{14}$ to $3 \times 10^{17}$ photons cm$^{-2}$ s$^{-1}$). Response reliability was strikingly high across trials ($n = 4$) for both current and firing rate activities (Fig. 4j). Together, these results demonstrate that RGCs expressing ChR-tdT can follow a high dynamic range of light stimulation compatible with human perception.

**ChR-tdT can generate a high spatial precision for visual perception.** Having shown that the RGC responses precisely follow

the temporal resolution of optogenetic stimuli we then tested the spatial sensitivity of optogenetic responses, by stimulating the retina on the MEA using circular spots of various sizes (25 µm, 50 µm, and 100 µm) centered on the MEA electrodes (100-µm electrode pitch, 10 µm diameter) at a light intensity of $2.10 \times 10^{17}$ photons cm$^{-2}$ s$^{-1}$ (600 nm ± 10 nm) (Fig. 5). The multiunit electrode-based analysis showed that even the electrodes far away (up to 1 mm) from the stimulated spot elicited an increase in spiking frequency (Fig. 5a). For identification of the electrode closest to the recorded cell, we performed spike sorting on the electrode signals, to obtain single-cell activity with an unsupervised sorting algorithm (Supplementary Fig. S4). This spike sorting indicated that individual spikes were recorded on several electrodes, as a consequence of spike propagation in the RGC axons running along the surface of the retina toward the optic disk (Fig. 5A and Supplementary Fig. S5). The increase in latency with distance to the stimulated area was consistent with an anterograde propagation of spikes along axons. We made use of this spike propagation to measure the spike velocity in ChR-tdT-expressing cells (Supplementary Fig. S5). The unimodal distribution peaking at 0.5 m/s (Supplementary Fig. S5H) suggests that the ChR-tdT-expressing population of RGCs contains a majority of midget RGCs[35]. This conclusion concerning cell identity is consistent with the midget cell morphology of tdTomato-expressing cells observed on two-photon microscopy (Supplementary Fig. S5A–E). However, a very small number of cells ($n = 9$) had faster velocities of axonal spike propagation (>1 m/s), indicating the possible presence of parasol RGCs among the ChR-tdT-expressing RGCs.

As tdTomato fluorescence was detected in RGC axons, we investigated whether light stimulation could elicit spikes directly in ChR-tdT-expressing axons, with anterograde and/or retrograde propagation. When a spot of light was centered on an electrode in contact with ChR-tdT-positive axons but not ChR-tdT-expressing soma, we observed no increase in spike activity in any neighboring or distant electrodes (Fig. 5b). Thus, the optical stimulation of ChR-tdT expressed in axons was not sufficient to trigger spikes. Indeed, a high degree of correspondence was found between the area containing cell bodies expressing TdTomato and the location of electrodes with optogenetic responsive cells (Fig. 5c). When spot size and presentation duration were varied, we observed single-cell activation for spots as small as 50 µm (Fig. 5d–f and Supplementary Movie S1). The number of responsive cells and their spiking frequencies depended on spot size and stimulus duration (Fig. 5g, h and Supplementary Fig. S6). It should be noted that our stimulations were centered on the opaque MEA electrodes (10 µm diameter), potentially greatly decreasing light intensity for the smallest spot (25 µm in diameter). Nevertheless, these observations are consistent with the notion that optogenetic stimulation can provide a high spatial resolution in RGC activation.

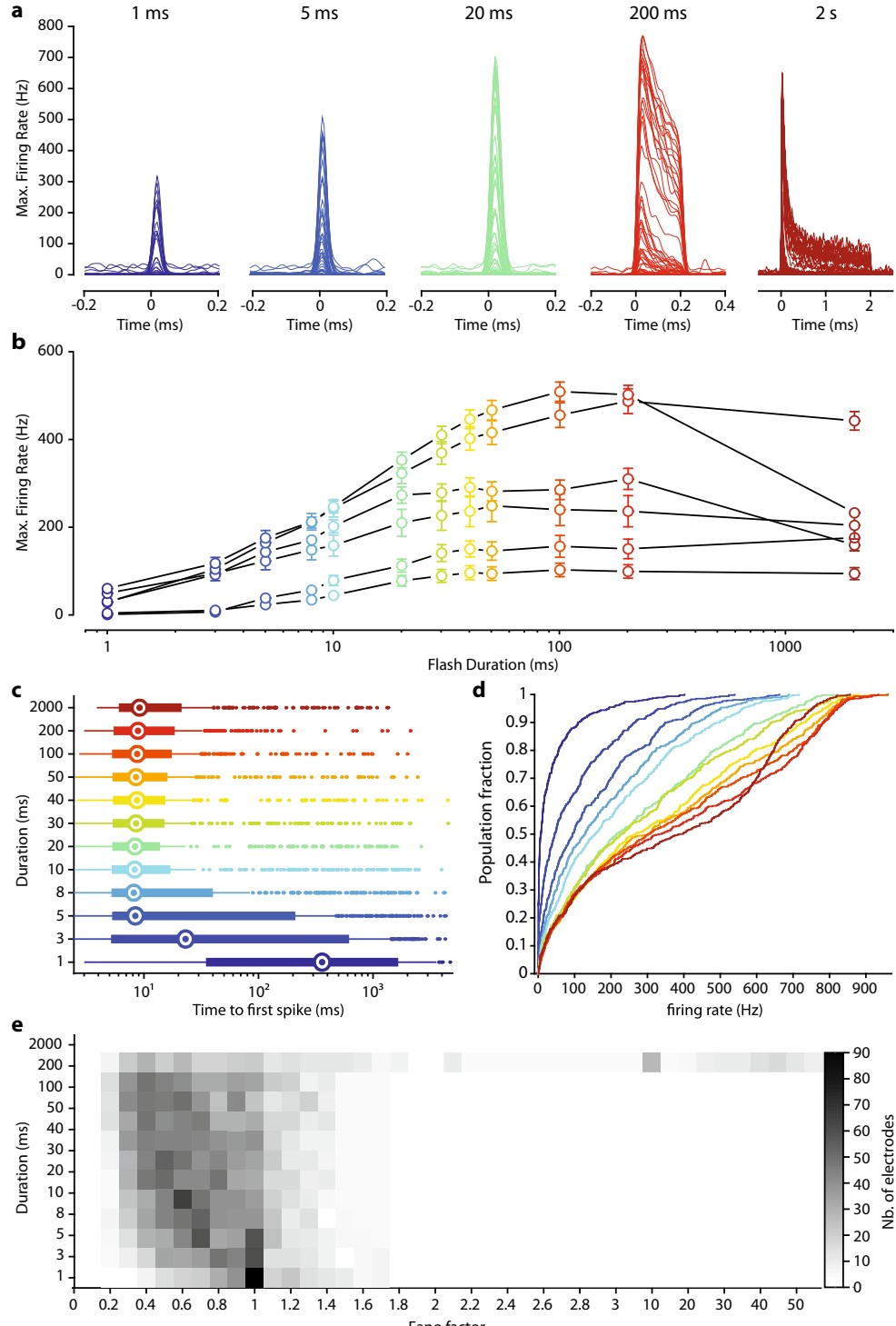

**Fig. 3 Millisecond activation of ChR-tdT-expressing primate RGCs. a** Spike density function for all responsive electrodes ($n = 66$) of one retina treated with for $5 \times 10^{11}$ vg in response to stimuli of increasing duration (1–5–20–200 ms, and 2 s, left to right with different colors) and constant light intensity ($2 \times 10^{17}$ photons cm$^{-2}$ s$^{-1}$, 600 nm ± 10 nm). **b** Mean maximal firing rate ± SEM measured for retinas treated with $5 \times 10^{11}$ vg/eye for all tested stimuli duration and constant light intensity ($n = 6$). **c** Horizontal box plot displaying time from the onset of stimulation to the first spike, as a function of stimulus duration. Recordings from the different retinas are pooled, such that each electrode has the same weighting. Medians are displayed as an open circle, box edges indicate the 25th and 75th percentiles, whiskers extend to the maximum and minimum, and outliers are plotted individually. **d** Cumulative plot of maximal firing rate per electrode versus stimulus duration, with duration color-coded as in **c**. **e** Distribution of Fano factor as a function of stimulation duration, for all responsive electrodes. A value of 1 corresponds to the Poisson distribution, and values below 1 indicate an increase in information content (**c**, **d**, **e**: $n = 488$ electrodes from six retinas expressing ChR-tdT).

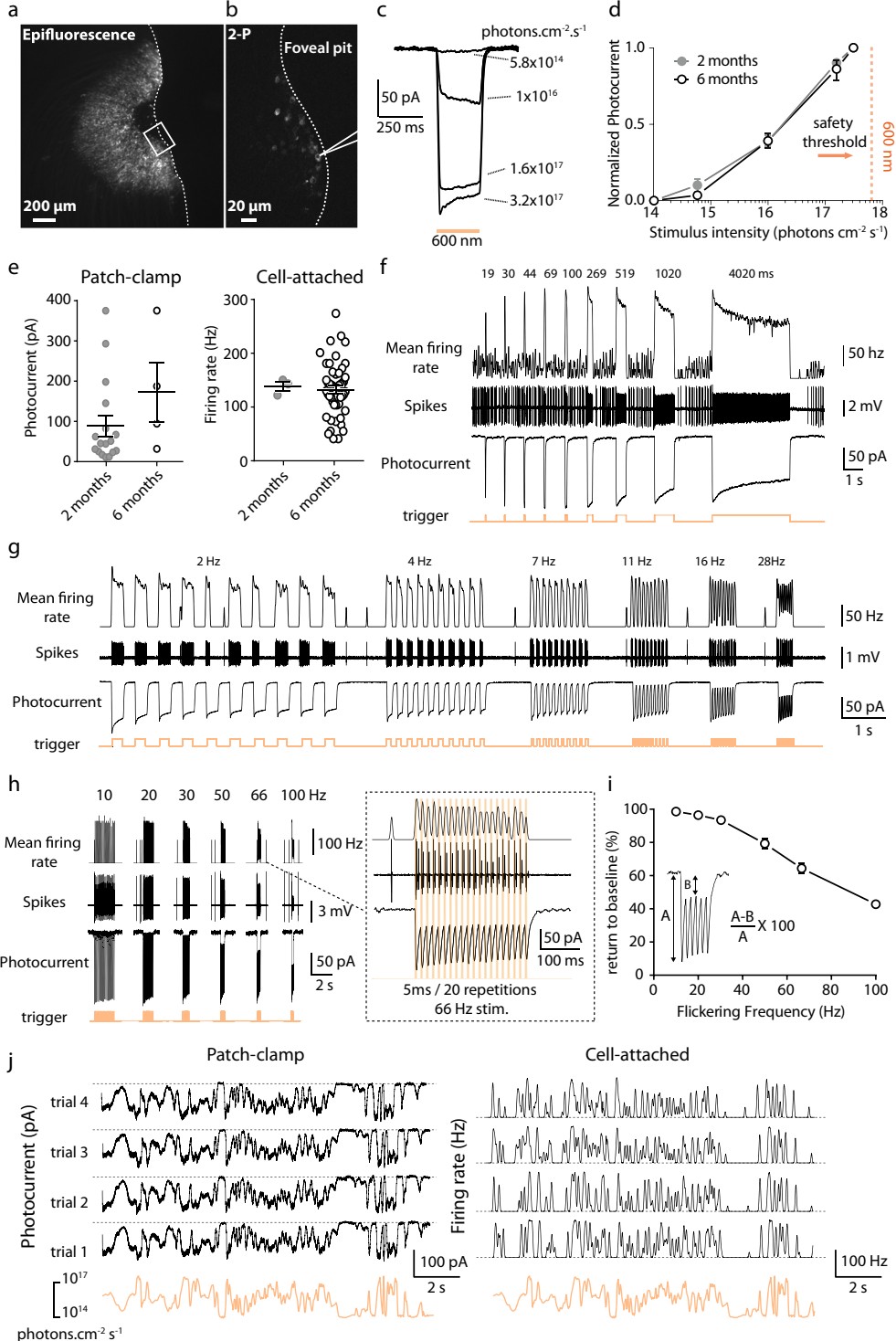

**The optogenetic stimulation of ChR-tdT-expressing RGCs can support pattern discrimination**. We assessed the functional impact of our visual restoration strategy by evaluating the ability of treated retinas to encode information about the direction and speed of motion and to discriminate patterns. We first presented moving bars (75 μm wide), at various velocities (2.2 mm/s or 4.4 mm/s), and with different orientations and directions across the treated retina (Supplementary Movie S2). Based on the known retinal magnification factor, 1 arc-degree of visual size corresponds to 211 μm on the retina[36]. Our bar stimuli, therefore, corresponded to a visual

field angle of 0.375° moving at 11 or 22 °/s. For calculation of the visual flow elicited, we used spike sorting on the recorded activity, followed by plane fitting to the peak of the cell responses, to estimate the direction of the bar (Fig. 6a and Supplementary Fig. S7) and its speed (Fig. 6d). The plane fitting method made it possible to identify, for each direction, the unique succession of cells activated along the path of the bar (Fig. 6a–c). This temporal response of the cells was found to be sufficient for correct estimation of the direction and velocity of the bar over the retina, despite the discrete spacing of electrodes.

**Fig. 4 Long-term stable expression of ChR-tdT allows reliable spike train generation at high temporal resolution. a** Epifluorescence image showing tdT-expressing RGCs in the perifoveal region of a monkey 6 months after injection of AAV2.7m8–ChR-tdT at $5 \times 10^{11}$ vg/eye. Limit of the retinal explants displayed as a dashed line. **b** Two-photon laser imaging of the foveal pit. The scanned zone is presented as a rectangle in **a**; the outline of the patch-clamp electrode is visible in the vicinity of a ChR-tdT-positive cell. **c** Photocurrent responses of a representative cell to different light intensities ($5.8 \times 10^{14}$ to $3.2 \times 10^{17}$ photons cm$^{-2}$ s$^{-1}$), obtained by patch-clamp recording. **d** Plot of mean normalized photocurrent ± SEM response against irradiance at 600 nm ± 10 nm in ChR-tdT-expressing RGCs following the injection of $5 \times 10^{11}$ vg/eye ($n = 17$ for 2 months, $n = 4$ for 6 months). The orange dashed vertical line indicates the safety threshold at 600 nm ($6 \times 10^{17}$ photons cm$^{-2}$ s$^{-1}$[28, 29]). **e** Peak photocurrent (left) and maximum firing rate (right) in ChR-tdT-expressing RGCs for the two durations of expression, 2 and 6 months (photocurrent: 88.7 ± 25.5 pA, $n = 17$ represented by closed gray circles and 172.4 ± 74.9 pA, $n = 4$ represented by open white circles, respectively, $P = 0.155$; firing rate: 138.3 ± 8.4 Hz, $n = 3$ and 132.3 ± 7.2 Hz, $n = 50$, respectively, $P = 0.669$). Long horizontal black lines indicate the mean ± SEM. **f**, top: Mean firing rate in response to stimuli of increasing flash duration (19–4020 ms) obtained in cell-attached configuration aligned with the trigger of the flash appearance (orange bottom line). **f**, middle: Raw data showing the recorded spikes in the cell-attached configuration. **f**, bottom: Photocurrents recorded in the same cell during the presentation of the same pattern of stimuli in the whole-cell configuration. **g** Firing frequency (top), spike trains (middle), and photocurrent response (bottom) of the same cell to increasing stimulus frequency in a full duty cycle (from 2 to 30 Hz). **h** left: Firing frequency (top), spike trains (middle), and photocurrent response (bottom) of the same cell to increasing stimulus frequency, for short light stimulation pulses (trains consisting of 20 repeats of a 5 ms stimulus), at 10–100 Hz. **h**, right: Close-up of the left panel showing responses to 5 ms stimuli presented at 66 Hz (20 repetitions). The RGC follows precisely the stimulus by firing distinct action potential doublets at every pulse. **i** Inactivation of the photocurrent, as observed in (**h**), as a function of stimulation frequency. For every cell recorded ($n = 7$), we measured the percentage return to the current baseline (averaged across the 20 stimulations), for stimuli of 2–100 Hz. For stimulations at 10, 66, and 100 Hz, cells returned to (mean ± SEM): 98.6 ± 0.4%, 64.4 ± 2.9 %, and 42.8 ± 1.8% of their initial baseline values, respectively. For panels **e–hi**, the light intensity used was $3.2 \times 10^{17}$ photons cm$^{-2}$ s$^{-1}$ at 600 nm ± 10 nm. **j** Representative example of the photocurrent response (left) and firing frequency (right) of a single cell, demonstrating high reliability across trials ($n = 4$) for both current and firing activity, during stimulation of the cell with a randomized oscillating stimulus intensity ranging from $3 \times 10^{14}$ to $3 \times 10^{17}$ photons cm$^{-2}$ s$^{-1}$ (orange lines). The dashed line illustrates 0 pA (left) or 0 Hz (right).

The most widely used clinical test for measuring visual acuity, the Snellen chart, assesses patient performance in reading letters and can be used to evaluate vision restoration strategies[37]. We estimated the visual acuity our strategy could be expected to restore, by stimulating retinas with different optotypes (X shape, circle, and square) of various sizes, from 55 to 330 µm (symbol width). All the characters presented were moved over the retina explants through the fovea center (8 different directions, 50 trials each, randomized presentations; Supplementary Movie S3), and the recorded spikes were sorted to analyze single-cell responses. Figure 7a represents the different patterns of cell activation for the moving shapes (top row: X, middle row: circle and bottom row: square) at a different time in their trajectories (from 12 ms to 114 ms, from the left to the right column). While the same cells can be recruited by different optotypes, the temporal pattern of cell activation depends on the shape presented (Fig. 7b). To confirm this, we used an algorithm[38] (see "Methods" for details) to discriminate directly between symbols of similar size from the spike responses generated by the ChR-tdT-expressing cells. We obtained for a symbol size of 220 µm a discrimination rate of 83% (Fig. 7c, the edge of the symbol: 44 µm), and a consistent maximal value for mutual information (Fig. 7d).

Due to the imperfections inherent to spike sorting and electrode array sampling, only a fraction of the transfected cells were detected (~80 cells after spike sorting vs. >1000 ChR-tdT-expressing cells per retina (Fig. 2)). Our estimates of the information transmission capabilities of this strategy are, thus, almost certainly considerably underestimated. Interestingly, for smaller stimuli, we measured a steady increase in discrimination accuracy and information (Fig. 7c, d). With access to the complete information transmitted by RGCs, we would probably have achieved a higher rate of discrimination for smaller symbols.

Despite this limitation, our findings demonstrate the ability of our strategy to encode information about the speed and direction of stimuli, even for small fast-moving stimuli, and its ability to support a discrimination task.

## Discussion

We show here that the AAV2.7m8–ChR-tdT construct is more efficient than the wild-type capsid AAV2 and the non-fused ChrimsonR protein in primate retinal ganglion cells. Furthermore, the therapeutic vector dose was defined as $5 \times 10^{11}$ vg/eye, which allows greater light sensitivity, with expression in more cells and over a wider area. We explored key parameters of optogenetic activation (i.e., light intensity, temporal, and spatial modulation) and demonstrated an ability to decode the direction and speed of a moving bar and the identity of different stimulus shapes from the recorded cell activity. These data supported the application to launch the ongoing clinical trial in patients with retinitis pigmentosa[39].

**Vector optimization for a high level of functional efficacy.** For the successful achievement of our therapeutic goals, the viral optogenetic construct must have a large functional impact on a large proportion of the cells in the RGC population. Our results confirm the higher transduction efficiency of the AAV2.7m8 variant in the retina[25] relative to AAV2[8,20]. Surprisingly, we also found that efficacy was greater for the ChR-tdT fusion protein than for ChR alone. We cannot entirely rule out an experimental bias, as ChR expression cannot be localized by fluorescence imaging as for ChR-tdT. However, the expected location of the transduced gene at the fovea[20,25] greatly decreases the chances of missing expression clusters during tissue isolation and positioning on the MEA. Differential protein trafficking, with td Tomato working as a trafficking helper and possibly also preventing protein aggregation, appears more likely[40]. The result would be a higher level of opsin construct targeting to the membrane[41]. We demonstrated a greater efficacy for both the mutated AAV capsid, AAV2.7m8, and for the ChR-tdT fusion protein, in non-human primates. Given the high degree of structural similarity between the eyes of NHPs and humans, we would expect intravitreal AAV2.7m8–ChR-tdT injection to transduce the retina effectively in blind patients too.

**Dose selection.** Once the construct has been selected, the next critical issue is AAV safety. Previous studies of gene therapy within the eye used AAV doses of $1.5 \times 10^{11}$ vg in the subretinal space[42] and of up to $1 \times 10^{11}$ vg/eye[43] or $1.8 \times 10^{11}$ vg/eye[44] for intravitreal injections. No adverse effects were reported at these doses, but other delivery methods use much higher doses (up to $10^{14}$ vg) with potentially disruptive effects[45]. In this study, none

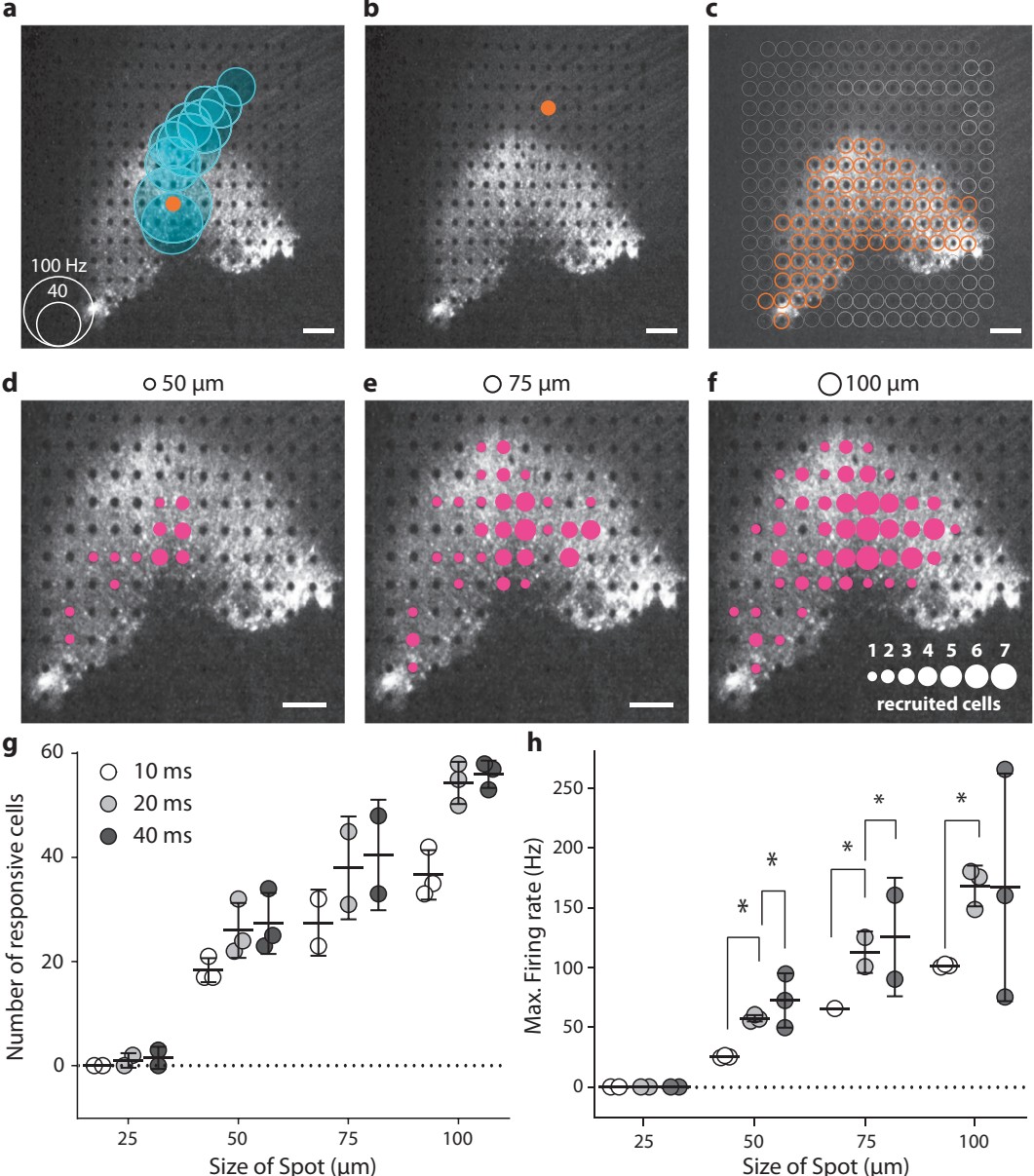

**Fig. 5 Spatiotemporal resolution for discrete stimulation. a** A focal light stimulation activated several different electrode channels on the transfected perifoveal area. The black and white image shows the fluorescence present in the perifovea area, with electrodes visible as black dots The orange dot shows the site of light stimulation (with a diameter of 100 μm and a duration of 40 ms spots. The blue circles are centered on the channels for which spikes were recorded suggesting a strong propagation activity. The size of the blue circles is proportional to the maximum spiking rate within a 100 ms window after stimulation, according to the scale in the bottom left corner. **b** When the light spot is directed toward another electrode at a location at which the RGC axons were stimulated, rather than the soma as in a; no response was measured (absence of blue circles). **c** Orange dots represented the sites at which stimulation resulted in a spiking response recorded on at least one MEA channel. These locations exactly match the pattern of intense fluorescence in the perifoveal area. **d–f** Images showing the extent of responses for 20 ms presentations of spot stimuli of different sizes (displayed at the top of the images). The sizes of the magenta circles indicate the numbers of cells activated by a spot at that particular position (see the scale in the lower right part of the panel). For **a–f**, scale bars indicate 200 μm. **g**, **h** Total number of recorded cells (**g**) and mean firing rate (**h**) over the entire electrode array is highly correlated with spot size and duration. For all stimulus durations (10 ms, 20 ms, and 40 ms), the increase in size from 25 to 50 μm, 50 to 75 μm, and 75 to 100 μm led to a significant increase in the firing rate of the cells, not shown on the figure (all $P < 0.005$, n = 3, see "Methods" for details); for increases in duration for single spot size, the results of statistical analyses are displayed on the figure.

of the eyes treated displayed an inflammatory response (Supplementary Fig. S1), with only a few cells in the vitreous and one eye displaying vitreal haze (associated with light hemorrhaging during the IVT procedure). Doses of $5 \times 10^{10}$ and $5 \times 10^{11}$ vg/eye induced ChR-tdT expression and strong functional responses highly efficiently in NHPs. The highest dose used here ($5 \times 10^{11}$ vg/eye) appears to provide more extensive retinal coverage and

higher light sensitivity (Fig. 2). This greater coverage would enlarge the patient's visual field, translating to ~6° in the visual field (211 μm on the primate retina per degree angle[36]). A visual field of this size may appear rather limited, but it should be borne in mind that the fovea is the center of high visual acuity. This high visual acuity results from the high density of cone photoreceptors, and the marked predominance of midget RGCs, which

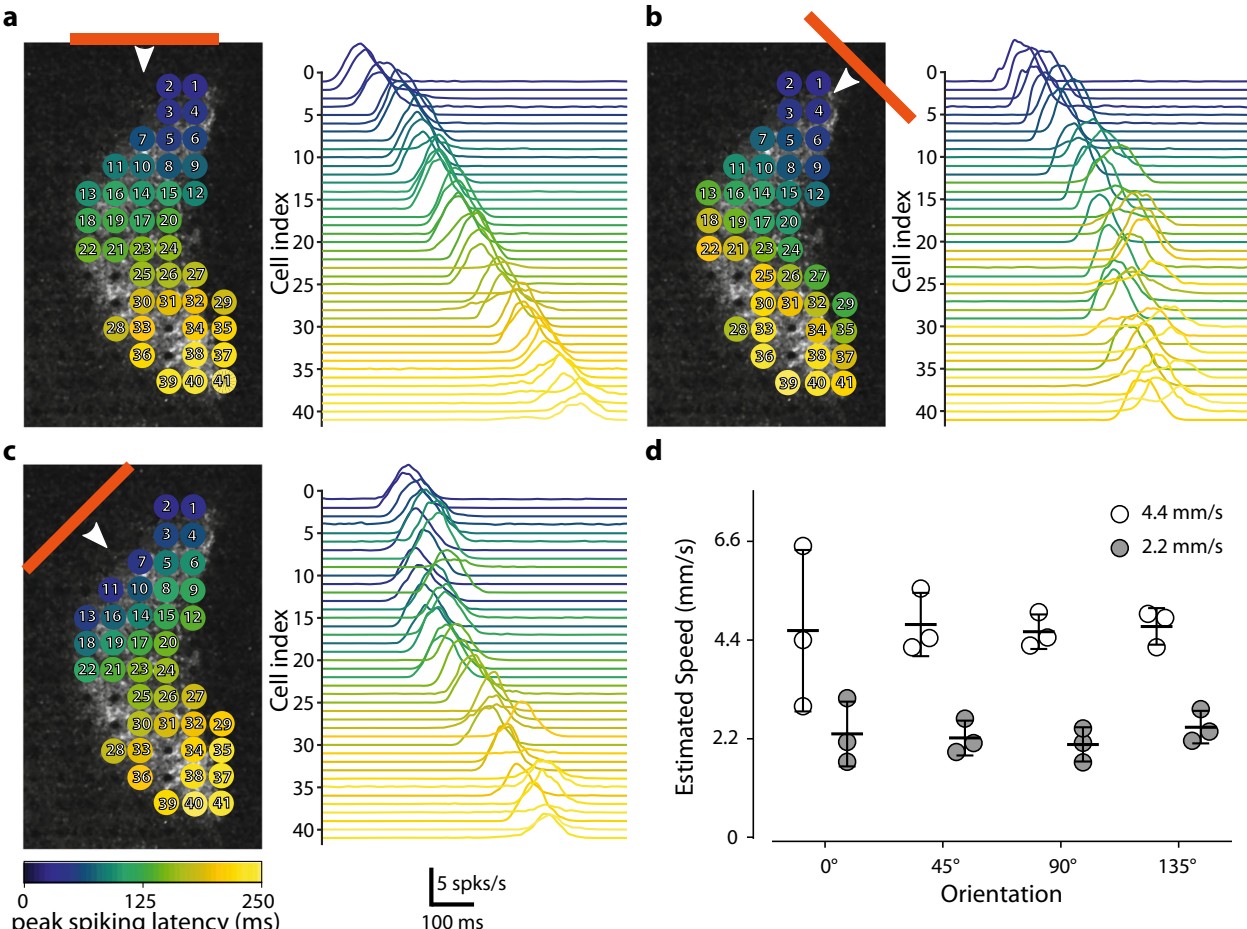

**Fig. 6 The measured activity is sufficient for estimation of the speed and direction of a moving bar. a** Various responses of the individual cells to a horizontal bar moving (75-μm long represented by the horizontal orange line moving at 2.2 mm/s downwards as indicated by the white arrow) across a transfected retina. The various cells recorded are represented in the panel on the left. The colors indicate the times at which peak spiking activity occurred for each cell relative to the earliest cell activation peak for the motion of the bar. The right panel shows the mean firing plots for all the recorded cells. The cells are ordered according to peak spiking latency. **b**, **c** As in **a** but with other two moving bar directions (45° for **b** and 135° for **c**). The cell index was retained. **d** A simple plane fitting-based method predicted the relative speed of the bars presented at either 2.2 mm/s or 4.4 mm/s ($n = 3$).

receive inputs from a single photoreceptor and have a very small receptive field[46–49]. We showed, by spike propagation speed analysis (Supplementary Fig. S5) and morphology examinations, that most of the ChR-tdT-expressing RGCs were, indeed, midget RGCs, likely to mediate visual perception with a high level of acuity. Specific activation of this midget RGC population in the foveal area could potentially provide patients with high-acuity vision.

**Effective and safe stimulation intensity.** We show here that the stimulation of ChR-tdT, as for most microbial opsins, is effective from $10^{15}$ photons cm$^{-2}$ s$^{-1}$. Given that only a quarter of visible light effectively stimulates any given light-sensitive channel, it would be hard to find situations in everyday life in which stimulation would occur. Outside on a bright day, the effective light intensity on the retina would be around $10^{14}$ photons cm$^{-2}$ s$^{-1}$, and this would fall to $2 \times 10^{12}$ photons cm$^{-2}$ s$^{-1}$ in an office. We would not, therefore, expect transfection with opsin to yield useful levels of perception in isolation. Our strategy will need to include an external photostimulation device for converting images into tailored patterned photostimulation of the optogenetically engineered retina.

This study focused on the red-shifted opsin ChrimsonR, which has a reported peak sensitivity at 575 nm[23], a result confirmed

here by MEA recordings (Fig. 1e). This is a much safer wavelength than highly phototoxic blue-light wavelengths, making it possible to expose the retina to higher light intensities safely[29]. For the clinical trial, we opted for a middle ground between optimal opsin sensitivity and the lower phototoxicity of higher wavelengths. We decided to use a 595 nm LED (Cree XP-E2, Lumitronix) as the light source for external photostimulation. For this reason, we mostly used light at a wavelength of 600 nm (±10 nm) in this study, and this is the wavelength considered for the safety evaluation.

The International Commission on Non-Ionizing Radiation Protection published limits for ocular exposure to visible and infrared radiation in 2013[28]. These limits were translated into retina irradiance by Sengupta et al., using published conversion rates[50,51]. The resulting threshold for continuous exposure was $5.6 \times 10^{17}$ photons cm$^{-2}$ s$^{-1}$ at 590 nm and $5 \times 10^{15}$ photons cm$^{-2}$ s$^{-1}$ at 500 nm. In 2016, Yan et al. published a review of the 2014 ANSI Z136.1 exposure limits for laser illumination of the retina with ophthalmic instruments (§8.3) and the potential use of these limits in optogenetic systems[21,32]. For a full-field continuous stimulus applied for 8 h during a 48-h period, the maximal permissive retinal peak irradiance is $1.1 \times 10^{17}$ photons cm$^{-2}$ s$^{-1}$ at 600 nm and $8 \times 10^{15}$ photons cm$^{-2}$ s$^{-1}$ at 505 nm.

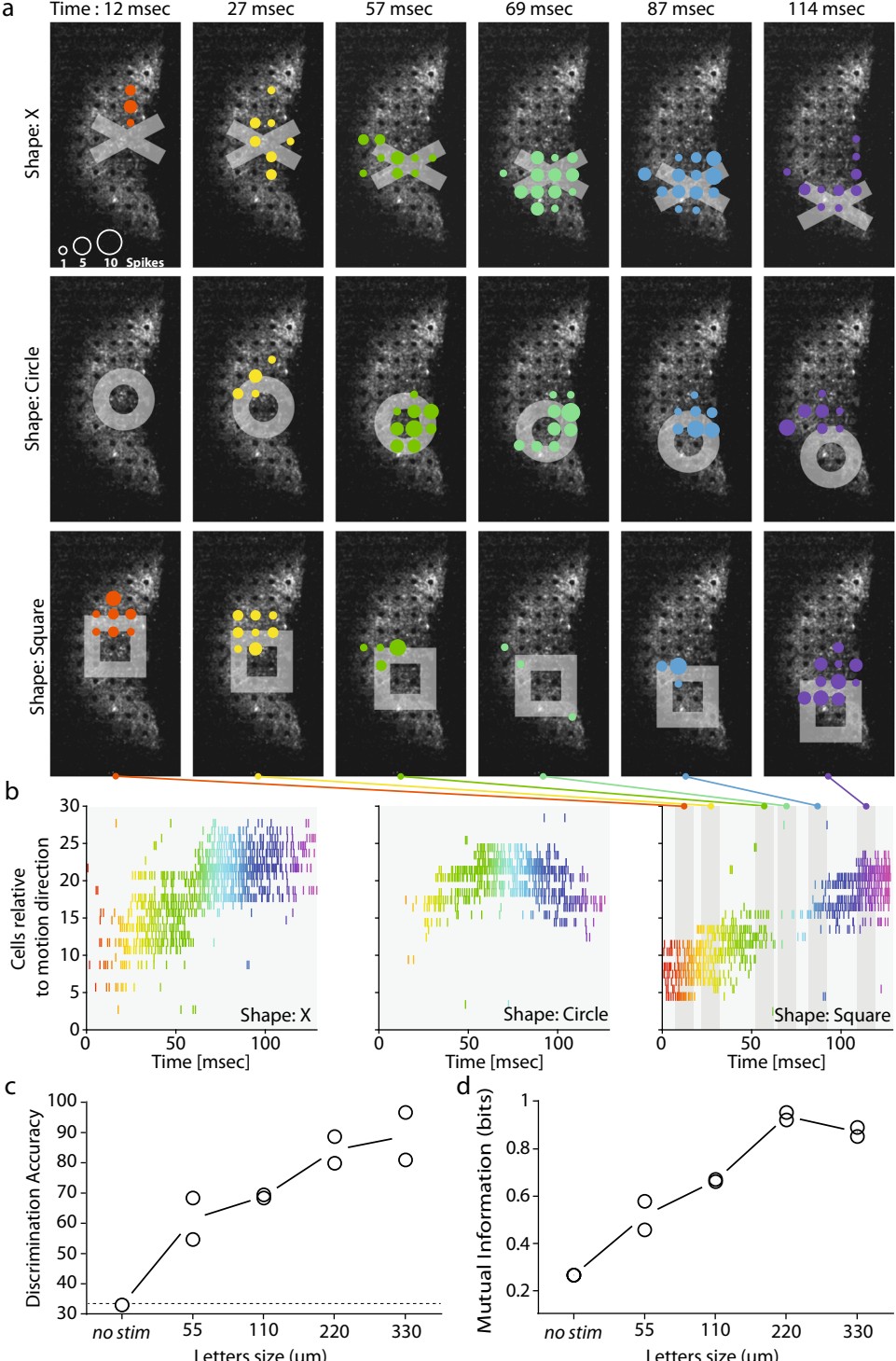

**Fig. 7 The measured activity can be used to discriminate stimulation with different shapes and to predict visual acuity. a** Responses of cells to three different shapes (rows: X, circle and square, columns: six different times) moving from the center towards the bottom of the retina. The colored disks indicate the electrodes where cells are recorded spiking. The size of the disks show the number of spikes detected in a 10-ms time window centered on the time indicated at the figure top. Color of the disks changes with timing in the sequence. **b** Raster plots showing the activity of the cells over time. The three shapes (X: left, circle: center and square: right) triggered different patterns of activity in the cell population. Color of the action potential on the raster change with time, fitting the color code in **a**. The six different time windows used in a are shown as the gray area over the right panel (square). **c** Discriminability (as **a** %) of the features of the spatiotemporal spike response obtained from population spike trains for different sizes. Discriminability increased with letter size. "No-stim" discrimination was performed with the spontaneous activity pattern just before the start of stimulation. The no-stim discrimination rate was about 33%, as would be expected by chance alone for three shapes. **d** Normalized mutual information from the confusion matrix obtained from the discrimination rates in (**c**). The information tends to increase with the size of the shapes, with saturation occurring at 220 μm. The information and discriminability at 220 μm indicate a visual acuity above the defined threshold for legal blindness.

Both sets of proposed exposure limits are higher for the 600-nm wavelength used here than for the blue light used for opsins such as Chr2. Importantly, we demonstrate here that light intensities below the 600 nm radiation safety limit can activate the transfected retina very efficiently.

Exposure limits are, by nature, conservative. They are set for continuous illumination, as the photochemical hazard, which depends on the total illumination received per 48 h, is the highest risk. However, our strategy will involve stimulation with light patterns extracted from an event camera, reducing moving faces and objects to their outlines[31,52]. The scarcity of stimulation will increase the maximal retinal peak irradiance permitted[21].

The risk of photochemical injury in the retina of patients with advanced RP lacking functional photoreceptors remains unclear, but the build-up of potentially toxic retinoids in the RPE would, presumably, be minimal.

For further guarantees of patient safety during the clinical trial, it will be important to limit the initial duration of exposure and to monitor the state of the retina after each stimulation period very closely.

**Comparison with other studies and constructs**. A previous clinical trial of optogenetic therapy for visual restoration focused on the blue-sensitive microbial opsin, Chr2[9]. This first clinical study built on preclinical studies in mice[7] and marmosets[8]. In the marmoset study, a single MEA electrode recorded spike trains reaching >300 Hz at $6.6 \times 10^{16}$ photons $cm^{-2} s^{-1}$. More recently, using the human codon-optimized $Ca^{2+}$-permeable Chr2, (CatCh), which is 70 times more efficient than Chr2, we recorded multiunit spiking frequencies in a similar range in macaque foveal RGCs (~300 Hz at $8 \times 10^{15}$ photons $cm^{-2} s^{-1}$ [20]). These studies focused on opsins sensitive to blue-light wavelengths and they reported results at intensities above safety limits. In this study, we observed multiunit spiking frequencies above 700 Hz (Fig. 3a) at $2 \times 10^{17}$ photons $cm^{-2} s^{-1}$ and a peak firing rate above 300 Hz for light intensities well below the safety limits ($9 \times 10^{15}$ photons $cm^{-2} s^{-1}$, Fig. 2c).

ChrimsonR is currently the most red-shifted opsin available, with peak sensitivities shifted by ~100 nm for Chr2 and 45 nm for ReaChR[23], but future work may even result in the development of infrared-sensitive opsins, as in snakes[53,54]. Alternatively, mutagenesis could be used to enhance the properties of existing opsins: ChrimsonR was generated by the site-directed mutagenesis of Chrimson[23], and was further modified to drive neuron firing rates to higher spiking frequencies[22]. This kinetic enhancement was achieved at the expense of light sensitivity, and this new variant is not, therefore, relevant for vision restoration, as it would reduce the safe range for stimulation. Based on previous studies aiming to develop an optogenetic vision restoration strategy, we show here a highest level of evoked activity at a wavelength of 600 nm, with a temporal resolution of milliseconds, at intensities below the safety threshold.

**Pattern discrimination for the restoration of vision**. The restoration of vision with retinal prostheses is classically assessed by stimulating individual electrodes[55,56], defining object positions and shape[5,56–58], identifying bar orientation[5,55,57], and reading letters or words. An issue encountered with epiretinal prostheses is that single-electrode stimulations activate RGCs axons on their way to the optic nerve, leading to patients perceiving an arc, rather than a point[59]. In our optogenetic approach, RGCs axons expressed ChR-tdT, as indicated by tdT fluorescence, but the light stimulation applied did not trigger spikes (Fig. 5b, c). This specificity of activation is a requirement for spatially restricted stimulation. We show here through spot stimulations that responses

can be elicited with a spot diameter of 50 μm, even for a duration of 10 ms (Fig. 5 and Supplementary Fig. S6). These findings are encouraging, but probably represent an underestimate of actual spatial resolution, for two reasons. First, the circular spot used in these experiments was centered on the 10 μm opaque electrodes, reducing photon flux, especially for smaller stimuli (i.e., 25 μm, Fig. 5). Second, due to the limitations on sampling during MEA recordings, it is unlikely that we recorded the activity of all responsive cells. It is, therefore, difficult to compare our results with those for subretinal implants, because most of these prosthetic devices activate RGCs indirectly through bipolar cells. However, electrodes have a 70–100-μm pitch that can be activated only in a stepwise manner, as all or most of the implanted electrodes must be stimulated for an activation current to be generated[5,60,61]. Whereas optogenetic therapy can activate RGCs with smaller spot size, and increasing the size of the stimulation spot increases the number of cells recruited in an almost linear manner (Fig. 5h).

Visual acuity is considered normal for a value of 20/20, corresponding to the ability to identify an object of 5 arc-minutes, with a critical gap of 1 arc-minute that needs to be resolved. The best reported visual acuity achieved with current retinal prostheses was 20/546, when assessed with Landolt C-rings[5]. In our optogenetic strategy, using an approach similar to the Snellen chart, we obtained correct shape discrimination (Fig. 7: 83% discrimination for symbols of 220 μm with 44-μm edges). For adult *Macaca fascicularis* monkeys, 1 arc-degree in the visual field corresponds to a size of 211 μm on the retina[36]; to obtain 20/20 acuity, a gap of 3.6 μm must be resolved,. In our case, the size of the gap correctly discriminated (edge of the symbol: 44 μm) represents a visual acuity of 20/249, which is above the legal threshold for blindness (20/400[62,63]). This value is consistent with the predictions of simulations[64] and better than any acuity reported to date with visual prosthetics (20/546[5]). The impossibility of recording all cells from the ChR-tdT-expressing population would clearly have underestimated visual acuity, which could probably be enhanced even further. One important perifoveal feature not taken into account here is the lateral displacement of the RGC cell body with respect to its receptive field. Indeed, to ensure minimal optical aberration for the light hitting the photoreceptor in the most central part of the fovea, other retinal layers are displaced centrifugally around the fovea pit. Due to this displacement, the direct stimulation of RGCs should occur through a corrected image of the visual field. Finally, degenerated retinas have a much lower signal-to-noise ratio than our pharmacologically isolated RCGs, due to abnormal circuit reorganization. Furthermore, RGCs in the degenerating retina have abnormally high spontaneous firing rates, which may have a large effect on their response to optogenetic stimulation, decreasing visual acuity in patients. However, a recent study has suggested treating this consequence of tissue reorganization with retinoic acid blockers[65].

We showed, by making use of the reliability of spike train generation, that pulse width modulation might be preferable to continuous illumination (Fig. 4). Evaluations in patients will define the best duration of stimulation, but we can infer, from our data, that RCGs are reliably activated in the 5–20 ms range (Figs. 3 and 4). We hypothesize that this ability to evoke high-frequency modulation would help (1) to reduce the total amount of light entering the eye, and (2) to maintain precise control over cell activity, potentially improving the outcome of this strategy.

**Conclusion**. We describe here the initial selection process for genetic content, vector serotype, and vector dose in an ambitious "two-prong" vision restoration strategy involving a biological

component and an external light stimulation device. While we could not assess thoroughly the behavioral impact of our strategy on sighted or blind macaques, the results for the biological component presented here provide all the essential information required for the design of the external light stimulation device necessary for the conversion of visual scenes into a stimulation pattern. Both the biological component and external light stimulation device are now in use, together, in the recently launched phase I/II clinical trial of AAV2.7m8–ChR-tdT[39].

## Methods

**Animals**. Experiments were performed on 18 male and female crab-eating macaques (*Macaca fascicularis*). The exact number of animals included in the different experiments is listed in Supplementary Table S1. All experiments were performed in accordance with the National Institutes of Health Guide for the Care and Use of Laboratory Animals. The protocol was approved by the local animal ethics committees (#044) and conducted in accordance with Directive 2010/63/EU of the European Parliament. Animals are housed in a room lit with fluorescent tubes which are filtered through diffuser panels above each housing unit to deliver a cool white light (4000 K), during a 12-h photoperiod. The maximum amount of light that could be measured, corresponding to an animal looking directly at the light in the upper part of its enclosure, is 585 μW/cm$^2$. This maximal possible exposure does not reach the light intensity threshold for activating ChrimsonR, measured here at $2.34 \times 10^{15}$ photons cm$^{-2}$ s$^{-1}$ (784 μW/cm$^2$, Figs. 1d and 2e). Furthermore, the light to which the animals are exposed in these housing conditions is a broad white light with a large spectrum that would be unable to activate ChrimsonR in an optimal manner, unlike the pure 595 nm orange light used in the study. We, therefore, expected to see no behavioral changes due to ChrimsonR activation in these normal housing conditions. We nevertheless continually monitored the behavior of the animals, by checking that their feeding habits were normal and evaluating general stress and behavior during enrichment periods. None of the treated animals showed any sign of stress due to photosensitivity. No noticeable changes in behavior (during feeding and enrichment) were observed after the surgery.

**Statistics and reproducibility**. All data shown in the figures are expressed as the mean ± standard deviation. $P$ values below 0.05 were considered significant. In tests of different genetic constructs, the fraction of active and responsive electrodes were compared between groups in Chi-squared contingency tests, followed by Fisher's exact test (Fig. 1c). For calculations of RGC density as a function of eccentricity (Fig. 2b), we used Tukey's multiple-comparison tests and compared $5 \times 10^{09}$ ($n = 3$) to $5 \times 10^{10}$ vg/eye ($n = 4$): $P = 7.6e\text{-}3$ at 175 μm, $P = 1e\text{-}4$ for 225 μm; $P < 1e\text{-}4$ from 275 to 375 μm, $P = 2e\text{-}4$ at 425 μm, and $P = 0.02$ at 475 μm. Comparing $5 \times 10^{11}$ ($n = 6$) to $5 \times 10^{10}$ vg/eye ($n = 4$), we obtained: $P = 2.6e\text{-}2$ at 475 μm and $P = 4.5e\text{-}2$ at 525 μm. For the percentage of electrodes responding at the different virus concentrations and the two expression times (Fig. 2d), we performed, for each pair of conditions, a nonparametric Mann–Whitney test and showed no significant differences ($n = 4$ for all condition except for $5 \times 10^{11}$ vg/eye at 6 months, where $n = 8$). The additional amount of firing for stimulations at different light levels, for retinas treated with three doses of vectors (Fig. 2e), was compared in Tukey's multiple-comparison test (see Supplementary Table S2), $n = 1$, 3, and 8 for $5 \times 10^9$, $5 \times 10^{10}$ and $5 \times 10^{11}$ vg/eye at 6 months, respectively. We compared photocurrent size in patch-clamp conditions and firing rate in cell-attached conditions for the two expression times (Fig. 4e), in nonparametric Mann–Whitney tests (n.s., $P = 0.155$ and $P = 0.669$, respectively). For spot stimulation (Fig. 5) we performed pairwise $t$ tests on spike rate data for the recorded retinas in different stimulation conditions. For stimulus of increasing size compared with the same duration: 25–50 μm, $P = 7.67e\text{-}7$, $P = 5.41e\text{-}8$, and $P = 5.41e\text{-}7$ for 10, 20, and 40 ms stimulation, respectively. In all, 50–75 μm, $P = 6.19e\text{-}8$, $P = 1.24e\text{-}6$, and $P = 6.40e\text{-}6$ for 10, 20, and 40 ms stimulation, respectively. In all, 75–100 μm, $P = 6.51e\text{-}8$, $P = 7.61e\text{-}6$, and $P = 1.6e\text{-}3$ for 10, 20, and 40 ms stimulation, respectively. Furthermore, we measured, when comparing the same stimulus size at a different time: 10–20 ms, $P = 7.942e\text{-}7$, $P = 0.0043$, and $P = 2.515e\text{-}6$, for stimulus size of 50, 75, and 100 μm, respectively. In all, 20–40 ms, $P = 0.0012$, $P = 0.0087$ for stimulus size of 50 and 75 μm, respectively. For Figs. 6 and 7, see Supplementary Methods for details.

**Reporting summary**. Further information on research design is available in the Nature Research Reporting Summary linked to this article.

## Data availability

The data that support the main findings of this study are openly available in figShare (https://figshare.com/projects/AAV2_7m8-ChrimsonR-tdTomato_for_vision_restoration/83675).

## Code availability

The codes supporting our findings are available on the repository Github (https://github.com/himstien/Optogenetic_Retinal_Data_Analysis).

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

## Acknowledgements

We thank Matthew Chalk for critical proofreading. We thank Guillaume Labernède and Antoine Rizkallah for the manual counts of RGCs. This work was supported by BPIfrance (grant reference 2014-PRSP-15), Gensight Biologics, Foundation Fighting Blindness, Fédération des Aveugles de France, and by French state funds managed by the Agence Nationale de la Recherche within the Investissements d'Avenir program, RHU LIGHT4DEAF (ANR-15-RHU-0001), LABEX LIFESENSES (ANR-10-LABX-65), IHU FOReSIGHT (ANR-18-IAHU-0001), and (ANR-11-IDEX-0004-02).

## Author contributions

D.P., P.H., J.A.S., D.D., J.D., R.B., and S.P. designed the study; A.D., M.D., and D.D. designed viral vectors; M.D. produced viral vectors; S.B. and E.B. performed IVT injection; C.M.F., J.D., C.J., and E.B. performed clinical follow-up on animals; P.P., F.A., and M.A.K. contributed to the in vivo tests; J.C. controlled all illumination devices; V.F. prepared the retina according to safety rules; G.G., H.A., R.C., and O.M. designed, performed, and analyzed MEA experiments; A.C. designed, performed, and analyzed patch-clamp experiments. G.G., H.A., and A.C. constructed the figures. G.G., H.A., A.C., F.A., J.C., P.P., and S.P. wrote the paper. All authors reviewed the paper.

## Competing interests

D.D., R.B., and S.P. are consultants for Gensight Biologics. J.A.S. and S.P. have financial interests in Gensight Biologics. D.P., A.D., D.D., D.J., R.C., G.G., M.D., J.A.S., and S.P. have filed a patent application relating to the results presented in this paper. The remaining authors declare no competing interests.
