## [Peer Review File · Communications Biology]

Reviewers' comments:

Reviewer #1 (Remarks to the Author):

This study demonstrates the efficacy of an optogenetic approach for generating visual sensitivity in retinal ganglion cells in non-human primate retinas. A microbial opsin, ChrimsonR (ChR), was introduced into retinal ganglion cells using an AAV2.7m8 viral construct. The viral construct conferred visual sensitivity to the ganglion cells, as demonstrated by light stimulation during pharmacological block of natural photoreceptor inputs. The data provide evidence for the efficacy, longevity and low-toxicity of the approach, which should be welcome news to researchers and clinicians who are currently using a similar AAV2.7m8-ChR-tdT construct in clinical trials (ref #37). Overall, the data is solid, and the presentation of the results is clear and convincing.

The data nicely confirm the utility of an optogenetic technology that has been developed and described across a number of studies in mice and non-human primates as reported by this and other groups (eg refs #19,20,22,24,29). Apart from the novel viral construct employed here, the techniques described appear to adhere to those developed previously. It would be helpful if the authors more clearly highlighted the novel outcomes, methods or mechanistic insights obtained from this study.

Specific comments:

1. The functional analysis of the spatial and temporal properties of the ChR-mediated light responses in the ganglion cells is potentially informative for interpreting the outcomes of the clinical trials. For example, visual acuity will depend on the spatial density of transfected cells, which can be estimated by resolving receptive fields of single cells. White-noise stimuli have been widely used to resolve the spatio-temporal receptive fields of single units in primate, mouse and other retinas, and therefore, it was disappointing that a similar analysis was not applied here. It was also surprising that the authors didn't take the opportunity to compare the sensitivity of the native, photoreceptor-driven light responses with those mediated by the ChR-tdT transfections in the same cells. Were there limitations imposed by the preparation that precluded collecting such data?

2. The success of the transfections were measured by counting the number of active cells, but as the authors point out, the same ganglion cell can be picked up on multiple electrodes. Was double counting significant, and if so were the data corrected?

3. A significant literature documents synaptic reorganization and the appearance of spontaneous noise within degenerated retinas. Phosphenes reported by affected patients are thought to reflect similar events. Such noise will be absent from the acutely isolated, pharmacologically blocked preparations used in this study. Perhaps the authors could comment on the potential for such spontaneous background noise to affect the signal-to-noise ratio of their ChR-mediated responses in degenerated retinas?

4. The assessment of transfection efficiency assumes complete suppression of the natural photoreceptor driven responses. Are there controls for the efficacy of the synaptic blockade of the photoreceptor light-responses? For example, recording from non-transfected regions before and after application of the blockers??

5. Fig. 5 is missing.

6. During patch recordings it seemed that only the ON-pathway was blocked using LAP4. Why were other antagonists omitted for these experiments?

7. The subheading, "AAV2.7m8 - ChR-tdT at 5x10¹¹ provides greater light sensitivity at 6 months post-injection", doesn't appear to agree with this statement in the results; "Comparing 2 and 6 months expression time, we do not observe any differences in the fraction of responsive electrodes ... nor in light sensitivity..."

Reviewer #2 (Remarks to the Author):

Review

Optogenetic therapy: High spatiotemporal resolution and pattern recognition compatible with vision restoration in non-human primates

The authors of this manuscript have presented a study on the effects newly designed AAV vectors and optimized opsin molecules in infecting retinal ganglion cells and leading to enhanced light sensitivity from less toxic wavelengths. They have carefully evaluated viral titer, immune response, photon sensitivity in a non-human primate, which possesses a fovea, and therefore closely resembles the human retina physiology. Using ex vivo imaging and recording they were also able to detect differences in spatial firing stimulated by distinct shapes suggesting this system is sensitive enough for object recognition on the millisecond timescale. This is a very promising study with a potential high impact and presents data that is being submitted to the FDA for a clinical trial. The scientific rigor is of sufficient quality. My enthusiasm is muted due to the many grammatical errors spread throughout the manuscript.

Minor points:

- 1) Not sure its fair to compare electrophysiology between those regions where you can identify based on fluorescence and those you cannot. It may well be that you are way off in your search. Typically when a construct doesn't have fluorescence we add GFP constructs so in that way one estimates the localization of the region successfully infected so and you have a rough idea where the potentially infected cells are located. This is what may have provided you with a way to compare. Without this, I am not sure it is fair to compare percent infected.
- 2) Bottom of page 7: "Taken together, these results confirmed the superiority of the AAV2.7m8 - ChR-tdT construct," Sorry, but you cannot draw this conclusion due to an unfair comparison. Please rewrite to more accurately reflect the obvious caveats.
- 3) The object recognition is not convincing. The neuron excitation in the video looks more specific to movement than a particular object. The excitation between the circle and square are remarkably similar and therefore to conclude these results imply shape discrimination is inaccurate and should be revised to more accurately reflect the results presented.

Reviewer #3 (Remarks to the Author):

I welcome this manuscript about the progress on vision restoration with optogenetic tools as not many clinical relevant results have been published about this topic. The authors themselves are involved in an approach by Gensight biologics. While the manuscript provides valuable information about optimal doses of viral vectors as well as the spatial resolution which can be obtained with this approach in an explanted retina, the ultimately interesting insights are not revealed, i.e. the behavioral impact. As the authors state "Our results here, on the biological component, provide all the essential information required for the design of the external light stimulation device necessary for the conversion of the visual scene into a stimulation pattern. Both parts are now in use, combined, in the phase I/II clinical trial that recently started using AAV2.7m8-ChR-tdT." This reads like there is much more interesting data which is not included in the manuscript. Of course, it would be much more interesting to see some behavioral data which I assume could have been generated with the non-human primates in the study (see point below). Nevertheless, the careful spatio-temporal characterization of optogenetic responses in NHP retina is interesting by itself and has not been shown in this detail in a scientific publication.

Based on photon estimations, the authors discuss that "it is hard to find any everyday life situation that would activate it". Nevertheless, it would be interesting to know whether the injected animals showed any signs of changes vision (e.g. increased light sensitivity or reduced visual capacity due to confusing/altered visual input). As the animals were housed for 2 -6 months, at least obvious

effects should have been documented. If no changes were observed this would be a valuable data point, too.

The authors claim that "Comparing 2 and 6 months expression time, we do not observe any differences in the fraction of responsive electrodes (Fig 2D)". While statistically this might have been the outcome, the figure itself might show a slight trend towards higher expression after the shorter expression time of 2 months. This might be due to fluctuations but on the other hand cells might be healthier after shorter expression times. To look into this aspect, close up images of expressing neurons (as in Fig. S3) as well as a comparison of densities of expressing cells as in Figure S1 would be helpful.

At some points the manuscript seems to have been written in a hurry. Some punctuations are wrong or missing, the manuscript contains typos, e.g. pg 5 "as ChR is often used fused to the fluorescent protein tdTomato", and a reference is not correctly listed (Prevot et al., 2019). Please double check the manuscript.

Reviewer #1 (Remarks to the Author):

This study demonstrates the efficacy of an optogenetic approach for generating visual sensitivity in retinal ganglion cells in non-human primate retinas. A microbial opsin, ChrimsonR (ChR), was introduced into retinal ganglion cells using an AAV2.7m8 viral construct. The viral construct conferred visual sensitivity to the ganglion cells, as demonstrated by light stimulation during pharmacological block of natural photoreceptor inputs. The data provide evidence for the efficacy, longevity and low-toxicity of the approach, which should be welcome news to researchers and clinicians who are currently using a similar AAV2.7m8-ChR-tdT construct in clinical trials (ref #37). Overall, the data is solid, and the presentation of the results is clear and convincing.

Authors' response:

All our thanks to the reviewer for the very positive comments: "data is solid" and "presentation of the results is clear and convincing"

The data nicely confirm the utility of an optogenetic technology that has been developed and described across a number of studies in mice and non-human primates as reported by this and other groups (eg refs #19,20,22,24,29). Apart from the novel viral construct employed here, the techniques described appear to adhere to those developed previously. It would be helpful if the authors more clearly highlighted the novel outcomes, methods or mechanistic insights obtained from this study.

Authors' response:

*As suggested by the reviewer, the novel outcomes, methods or mechanistic insights obtained from this study were underlined in the discussion (**Comparison with other studies and constructs** and **Pattern recognition for visual restoration**)*

Specific comments:

1. The functional analysis of the spatial and temporal properties of the ChR-mediated light responses in the ganglion cells is potentially informative for interpreting the outcomes of the clinical trials. For example, visual acuity will depend on the spatial density of transfected cells, which can be estimated by resolving receptive fields of single cells. White-noise stimuli have been widely used to resolve the spatio-temporal receptive fields of single units in primate, mouse and other retinas, and therefore, it was disappointing that a similar analysis was not applied here. It was also surprising that the authors didn't take the opportunity to compare the sensitivity of the native, photoreceptor-driven light

responses with those mediated by the ChR-tdT transfections in the same cells. Were there limitations imposed by the preparation that precluded collecting such data?

Authors' response:

All our recordings were performed on ex-vivo primate retinas, used 24 to 72h postmortem. After eye transportation in CO2 independent medium (at least 2-3h following euthanasia), dissection and removal of the RPE, the retina is chopped into pieces and conserved as retinal explants in a cell-culture incubator. This protocol, developed for post-mortem human and primate retinal recording, was previously described before in Sengupta et al. 2016, Chaffiol et al. 2017 and Khabou et al. 2019. This method presents the advantage to let the retina recover from hypoxia after a long transport reducing variability in the functional state of the retina. Furthermore, we decided not to dissect the macula attached to the RPE to reduce the risk of damaging this precious transfected area. Previous studies recording natural responses with RPE-attached retina on MEA were achieved on the peripheral primate retina not on the macula (Cafaro and Rieke, 2013, J. Neurosci, Li et al. 2015, J. Neurosci, Rhoades et al neuron 2019). In our conditions, most, if not all, natural photoreceptor responses were abolished and did not recover. As a consequence, we were not able to record natural light responses to compare them to optogenetic responses. To further verify that all natural responses were suppressed, we applied the pharmacological blockers to restrict recorded activity in the ganglions cells to intrinsic optogenetic responses.

We have included in the revised manuscript the fact that in our conditions, we were not recording natural responses.

*In **Results, subhead 1**, p. 6 line 8:*

“No natural light responses were recorded in our experimental conditions, but we nevertheless added synaptic blockers to the bath to suppress any residual natural light responses (see supplementary materials and methods).”

*As well as in **Supplementary Materials, Supplementary Materials and methods, Gene delivery, retina isolation and preservation of the primate retina**, p. 2, line 1 :*

“In these conditions, natural photoreceptor responses were abolished, and did not recover. We applied pharmacological blockers to check that all natural responses were suppressed (see below).”

The reviewer made an excellent remark on the use of white noise stimuli to best estimate the receptor field of our cells. We did perform similar protocols on some of our experiments, to use those receptor fields to add to the visual acuity estimation as in a previous study on optogenetic therapy (see Ferrari et al. 2018). However, the greater expression level with ChrR-tdT combined to the high light intensity and the high number (50%) of active pixels resulted in a rundown of the efficiency of responses as seen for 2 sec continuous stimuli (see fig. 3). As a consequence, we instead decided to measure visual acuity using an alternative approach relying on shape discrimination (Fig. 7). The observed rundown of activity is not a concern as stimulations through the bio-mimetic device is delivering short duration stimuli (up to few hundreds of milliseconds) preventing thereby the likelihood of long-term RGC illumination.

*We have included this difficulty to measure responses to white noise stimulation as a consequence of the activity rundown in **Results, Subhead 3**, p. 16, line 2:*

“...we observed a large increase in spike train variability for stimulations lasting 2 seconds (Fig. 3E). Most of this effect can be attributed to stimulus hysteresis, as retinal sensitivity subsequently recovered. Consistent with this observation, recordings of activity in response to achromatic binary white noise with a 50% pseudorandom selection rate revealed a gradual decline of evoked activity (data not shown). However, this suggests that limiting optogenetic stimulation to temporally discrete stimuli should be recommended, to generate strong and reliable cell responses.”

2. The success of the transfections were measured by counting the number of active cells, but as the authors point out, the same ganglion cell can be picked up on multiple electrodes. Was double counting significant, and if so were the data corrected?

Authors' response:

We did not perform spike-sorting on the data from figure 1 to 3, as cells responded to full-field stimuli and as such presented highly correlated activity, with recorded instantaneous firing rate in the order of 400 to 800 Hz, something that algorithms will have difficulties sorting correctly. Rather than displaying an imperfect cell sorting, we decided to represent multi-unit activity for full-field stimuli, which was obviously introducing multiple counting in all studied cases. In fact, this multiple counting does not change the result but it likely amplifies the difference among retina.

We added a sentence in **Results, Subhead 1**, p. 6, line 8, to reflect this choice:

“ No natural light responses were recorded in our experimental conditions, but we nevertheless added synaptic blockers to the bath to suppress any residual natural light responses (see supplementary materials and methods). The results shown are the multi-unit activity on all electrodes following full-field stimuli for the quantitative measurement of functional efficacy, which may have amplified the differences between results (see below). 256-MEA recordings revealed large differences between vectors in the ability to generate functional ChR expression (Fig. 1A-D).”

When we used more discreet stimulation (fig 5 to 7), we did perform spike sorting, and did observe that individual cell activity could be recorded on multiple electrodes. To illustrate this we included a supplementary figure describing differences in electrodes recordings multi-unit activity and the electrodes closest to the spike-sorted cells (**fig. S4**).

Fig. S4. Multi-unit activity detection versus position of the cell identified with a spike-sorting algorithm

(A-B) Recording examples for retinas treated with high dose, yielding high functional efficacy (A) or moderate functional efficacy (B). Open orange circles represent electrodes at which at least one neuron was identified by spike sorting, whereas filled circles represent electrodes at which multi-unit spiking activity from neurons located at the open circles was detected (threshold of 4x the SD of the signal before stimulation). Scale bar: 200 μm .

The text was modified accordingly, **Results, Subhead 5**, p. 23, line 15:

“Multi-unit electrode-based analysis showed that even the electrodes far away (up to 1 mm) from the stimulated spot elicited an increase in spiking frequency (Fig. 5A). For identification of the electrode closest to the recorded cell, we performed spike sorting on the electrode signals, to obtain single-cell activity with an unsupervised sorting algorithm (Fig. S4). This spike sorting indicated that individual spikes were recorded on several electrodes, as a consequence of spike propagation in the RGC axons running along the surface of the retina toward the optic disk (Fig. 5A & Fig. S5).”

Reviewer #1:

3. A significant literature documents synaptic reorganization and the appearance of spontaneous noise within degenerated retinas. Phosphenes reported by affected patients are thought to reflect similar events. Such noise will be absent from the acutely isolated, pharmacologically blocked preparations used in this study. Perhaps the authors could comment on the potential for such spontaneous background noise to affect the signal-to-noise ratio of their ChR-mediated responses in degenerated retinas?

Authors' response:

This is indeed an excellent point to raise, as we pharmacologically isolate the ganglion cells population the noise level is greatly reduced. Large levels of network reorganisation in the degenerated retinas would not particularly affect our strategy as we stimulate the output neurons of the retina. On the other hand, the increase of noise due to the reorganisation of the network could indeed affect the signal-to-noise ratio, and possibly decrease the visual acuity of patients. If this happens to be an issue, treatment could be developed to reduce background noise in the degenerated retinas as proposed by Teliás et al. with retinoic acid treatment.

We modified the **Discussion; Pattern recognition for visual restoration**, p.37, line 12, to add this point :

“Due to this displacement, the direct stimulation of RGCs should occur through a corrected image of the visual field. Finally, degenerated retinas have a much lower signal-to-noise ratio than our pharmacologically isolated RGCs, due to abnormal circuit reorganization. Furthermore, RGCs in the degenerating retina have abnormally high spontaneous firing rates, which may have a large effect on their

response to optogenetic stimulation, decreasing visual acuity in patients. However, a recent study has suggested treating this consequence of tissue reorganization with retinoic acid blockers (Telias et al. 2019)

Reviewer #1:

4. The assessment of transfection efficiency assumes complete suppression of the natural photoreceptor driven responses. Are there controls for the efficacy of the synaptic blockade of the photoreceptor light-responses? For example, recording from non-transfected regions before and after application of the blockers?

Authors' response:

Pharmacological blockers used in our MEA experiments, while they do suppress any remaining photoreceptor mediated light responses, also constrains the origin of the measured activation to the RGCs. This particular combination of antagonists was previously used in Chaffiol et al. 2018. For each experiment, we first recorded light responses without the blockers with lower light intensities than those required for the optogenetic activation. During these recording, we never observed any light response. On rodent models, we confirmed efficacy of our blockers in suppressing the natural light responses. The absence of any light response in many retina despite a high number of active RGCs confirms absence of natural responses. Furthermore, the recorded spectral sensitivity closely matches ChrimsonR known sensitivity (Fig1-E) suggesting further no natural photoreceptor-elicited response was preserved in our preparation.

We added in the text, **Results, Subhead 1**, p. 6, line 8:

“No natural light responses were recorded in our experimental conditions, but we nevertheless added synaptic blockers to the bath to suppress any residual natural light responses (see supplementary materials and methods).”

Reviewer #1:

5. Fig. 5 is missing.

Authors' response:

We are greatly sorry for this, it is now added to the manuscript.

Reviewer #1:

6. During patch recordings it seemed that only the ON-pathway was blocked using LAP4. Why were other antagonists omitted for these experiments?

Authors' response:

In the patch-clamp experiments, recording were first made in the absence of synaptic blockers. Single-cell recordings experiments showed that none of the tdTomato negative cells displayed light responses (data not shown, n>50). By contrast we systematically recorded ON-responses with fast photocurrents kinetics displaying microbial opsin fast activation signature in tdTomato positive cells. To further control that these ON-responses were strictly originating from ChrimsonR and not from potential residual functional photoreceptors, we added the on-pathway blocker LAP4 in the recording medium.

*We added a sentence in **Supplementary Materials, Supplementary Materials and methods, Two-photon live Imaging and single-cell electrophysiological recordings**, p2, line 23:*

“All the recorded responses were ON responses. We checked that these ON responses originated strictly from ChrimsonR, by making recordings in the presence of an ON-pathway blocker, the selective group III metabotropic glutamate receptor antagonist, L-(+)-2-amino-4-phosphonobutyric acid (L-AP4, 50 μM, Tocris Bioscience, Bristol, UK).”

Reviewer #1:

7. The subheading, “AAV2.7m8 - ChR-tdT at 5x10¹¹ provides greater light sensitivity at 6 months post-injection”, doesn't appear to agree with this statement in the results; “Comparing 2 and 6 months expression time, we do not observe any differences in the fraction of responsive electrodes ... nor in light sensitivity...”

Authors' response:

The subheading title has been changed, we do not observe differences between 2 and 6 months for the high dose, but high virus concentration vs medium and low dose does show a greater light sensitivity.

subhead 2, p. 10 changed to:

Subhead 2: AAV2.7m8 - ChR-tdT provides greater light sensitivity at a dose of 5x10¹¹ vg/eye

Reviewer #2 (Remarks to the Author):

Review

Optogenetic therapy: High spatiotemporal resolution and pattern recognition compatible with vision restoration in non-human primates

The authors of this manuscript have presented a study on the effects newly designed AAV vectors and optimized opsin molecules in infecting retinal ganglion cells and leading to enhanced light sensitivity from less toxic wavelengths. They have carefully evaluated viral titer, immune response, photon sensitivity in a non-human primate, which possesses a fovea, and therefore closely resembles the human retina physiology. Using ex vivo imaging and recording they were also able to detect differences in spatial firing stimulated by distinct shapes suggesting this system is sensitive enough for object recognition on the millisecond timescale. This is a very promising study with a potential high impact and presents data that is being submitted to the FDA for a clinical trial. The scientific rigor is of sufficient quality. My enthusiasm is muted due to the many grammatical errors spread throughout the manuscript.

Authors' response:

As suggested by the reviewer, the paper was edited by a professional English Editor.

Reviewer #2:

Minor points:

1) Not sure its fair to compare electrophysiology between those regions where you can identify based on fluorescence and those you cannot. It may well be that you are way off in your search. Typically when a construct doesn't have fluorescence we add GFP constructs so in that way one estimates the localization of the region successfully infected so and you have a rough idea where the potentially infected cells are located. This is what may have provided you with a way to compare. Without this, I am not sure it is fair to compare percent infected.

Authors' response:

While the reviewer raised a fair point, our goal was to evaluate clinical potency and adding another virus, to provide GFP labelling, would have skewed this goal (by putting AAV in competition). As outline in the beginning of results Subhead 1, multiple studies established that successful intravitreal injection in non-human primates leads to expression systematically in the peri-fovea region (Yin et al. 2011, Dalkara et al. 2013, Chaffiol et al. 2017). This very specific area is easily identified on the isolated retina and prepared for recording. The foveal area was extensively covered with the MEA recordings, and we changed the positioning of the retina explant multiple times when no light response was detected. Finally, even when td-tomato was expressed we did not use fluorescence to guide the positioning of the tissue on the electrode array but only to confirm correlation between recorded activity and td-tomato expression. While it's true that patch-clamp experiments were highly favorable to the genetic constructs with fluorescence, we made considerable efforts to reduce bias in multielectrode recordings.

*To take into account this comment, we changed two sentences in **Results, Subhead 1**, p. 6, line 18, to better describe this process:*

"This quantification revealed the existence of a significant difference between the constructs, with the highest efficacy for AAV2.7m8-ChR-tdT (Fig. 1C, 64.4% of active sites responsive vs. 13.4%, 10.6% and 0% for AAV2.7m8 – ChR-tdT, AAV2.7m8 – ChR, AAV2 – ChR-tdT and AAV2 – ChR, respectively, $P < 0.001$). For all constructs, the foveal area was identified and selected for recording. The

corresponding retinal explant was positioned on the MEA before confirmation of the eventual presence of fluorescence. If no light response was measured, we repositioned the tissue on the array to increase the sampling area. Light sensitivity was measured with a range of light intensities, from 1.37×10^{14} to 6.78×10^{16} photons.cm².s⁻¹ on all responsive retinas (Fig. 1B, Fig. 1D). Responses were obtained with AAV2.7m8 – ChR-tdT, in all four retinas tested (versus 2, 1 and 0 for AAV2.7m8 – ChR, AAV2 – ChR-tdT and AAV2 – ChR, respectively). “

Reviewer #2:

2) Bottom of page 7: “Taken together, these results confirmed the superiority of the AAV2.7m8 - ChR-tdT construct,” Sorry, but you cannot draw this conclusion due to an unfair comparison. Please rewrite to more accurately reflect the obvious caveats.

Authors' response:

As required by the reviewer, we tone down the conclusion **Subhead 1, Results**, p.7, line 21, to :

“In the absence of tdT fluorescence, for AAV2.7m8 - ChR and AAV2 – ChR, recordings were performed on random healthy RGCs in the perifoveal area (>40 cells per condition). In these conditions, none of the RGCs for which recordings were made displayed light-evoked responses, even under conditions known to activate ChR. We cannot exclude a potential bias in favor of the construct including tdTomato, particularly in the patch clamp experiments, but the positioning of the MEA on the basis of foveal identification would tend to exclude the possibility of such a bias in MEA recordings. These MEA recordings were consistent with a greater efficacy of the AAV2.7m8 - ChR-tdT construct, which was, therefore, systematically used in subsequent experiments.”

Reviewer #2:

3) The object recognition is not convincing. The neuron excitation in the video looks more specific to movement than a particular object. The excitation between the circle and square are remarkably similar and therefore to conclude these results imply shape discrimination is inaccurate and should be revised to more accurately reflect the results presented.

Authors' response:

As suggested by the reviewer, we tone down the conclusion on the object recognition: all occurrence of “pattern recognition” has been changed to pattern discrimination (including in the article title)

Although the spiking activity at electrodes may look similar during the pattern movement, spike sorting indicated strong differences in the cell activity pattern. Unfortunately, videos are representing the summed firing rate for the sorted cells reducing thereby the time precision for individual cell spiking. However, different shapes elicited different patterns of activity in individual cells allowing us to clearly differentiate activity patterns for different symbols at a given symbol size.

Reviewer #3 (Remarks to the Author):

Referee #3: circuit-based therapies, optogenetics

I welcome this manuscript about the progress on vision restoration with optogenetic tools as not many clinical relevant results have been published about this topic. The authors themselves are involved in an approach by Gensight biologics. While the manuscript provides valuable information about optimal doses of viral vectors as well as the spatial resolution which can be obtained with this approach in an explanted retina, the ultimately interesting insights are not revealed, i.e. the behavioral impact. As the authors state “Our results here, on the biological component, provide all the essential information required for the design of the external light stimulation device necessary for the conversion of the visual scene into a stimulation pattern. Both parts are now in use, combined, in the phase I/II clinical trial that recently started using AAV2.7m8-ChR-tdT.” This reads like there is much more interesting data which is not included in the manuscript. Of course, it would be much more interesting to see some behavioral data which I assume could have been generated with the non-human primates in the study (see point below). Nevertheless, the careful spatio-temporal characterization of optogenetic responses in NHP retina is interesting by itself and has not been shown in this detail in a scientific publication. Based on photon estimations, the authors discuss that “it is hard to find any everyday life situation that would activate it”. Nevertheless, it would be interesting to know whether the injected animals showed any signs of changes vision (e.g. increased light sensitivity or reduced visual capacity due to confusing/altered visual input). As the animals were housed for 2 -6 months, at least obvious effects should have been documented. If no changes were observed this would be a valuable data point, too. The authors claim that “Comparing

2 and 6 months expression time, we do not observe any differences in the fraction of responsive electrodes (Fig 2D)". While statistically this might have been the outcome, the figure itself might show a slight trend towards higher expression after the shorter expression time of 2 months. This might be due to fluctuations but on the other hand cells might be healthier after shorter expression times. To look into this aspect, close up images of expressing neurons (as in Fig. S3) as well as a comparison of densities of expressing cells as in Figure S1 would be helpful.

At some points the manuscript seems to have been written in a hurry. Some punctuations are wrong or missing, the manuscript contains typos, e.g. pg 5 "as ChR is often used fused to the fluorescent protein tdTomato", and a reference is not correctly listed (Prevot et al., 2019). Please double check the manuscript.

Reviewer #3:

Point 1;

Nevertheless, it would be interesting to know whether the injected animals showed any signs of changes vision (e.g. increased light sensitivity or reduced visual capacity due to confusing/altered visual input). As the animals were housed for 2 -6 months, at least obvious effects should have been documented. If no changes were observed this would be a valuable data point, too.

Authors' response:

*As suggested by the reviewer, we included information on the animal behavior in **Supplementary Materials, Supplementary Materials and methods, in Gene delivery, retina isolation and preservation of the primate retina, page 1, line 16:***

"None of the treated animals displayed any signs of photophobia or vision-related changes in behavior."

Reviewer #3:

Point 2

To look into this aspect, close up images of expressing neurons (as in Fig. S3) as well as a comparison of densities of expressing cells as in Figure S1 would be helpful.

Authors' response:

As advised by the reviewer, we examined further the cells at 2 and 6 months (see newly added figure S3). Although we cannot exclude the possibility of a decrease in the

number of ChR-tdTomato expressing cells between 2 and 6 months, no change in expression level was detected.

*We added the following sentence in the **result section, Subhead 2**, page 12, line 2:*

“We cannot exclude the possibility of a decrease in the number of ChR-tdT-expressing cells between two and six months, but no change in the level of expression was detected (Fig. S3). Furthermore, we observed no difference in the fraction of responsive electrodes (Fig. 2D, 102 ± 58 vs. 73 ± 65 for 2 months and 6 months, respectively, for 5×10^{11} vg/eye), or light sensitivity (Fig. 1D and Fig. 2E).”

A

2 MONTHS POST-INJECTION

CONFOCAL IMAGING**B**

6 MONTHS POST-INJECTION

**C**

2 MONTHS POST-INJECTION

LIVE FLUORESCENCE IMAGING**D**

6 MONTHS POST-INJECTION

Fig. S3. Morphological comparison at 2 and 6 months after AAV2.7m8-ChR-tdT treatment

(A-B) Confocal stack projections comparing ChR-tdT expression in the fovea of animals at 2 months (A) or 6 months (B) after intravitreal injection. Red: ChR-tdT. Blue: DAPI nuclear labeling.

(C-D) Live imaging, by epifluorescence (left images) or two-photon imaging (middle and right images), comparing ChR-tdT expression in the foveae of animals at 2 months (A) or 6 months (B) after intravitreal injections.

Reviewers' comments:

Reviewer #1 (Remarks to the Author):

My comments have been addressed, however, the response to the white-noise receptive field mapping does raise an interesting and pertinent question that was not immediately obvious in my initial reading. The authors note that white-noise analysis was not workable due to "rundown of the efficiency of responses" due to the continuous relatively high mean illumination level. This rundown doesn't seem to be evident in the photocurrents (Fig. 1G, 4C), suggesting that it likely arises from inactivation of voltage-gated channels in the ganglion cells. There seems to be some cell-to-cell variability (e.g. Fig. 1B, 4F,G), although the data in Fig. 3A suggests that on average spike-accommodation during high continuous illumination is strong. This makes sense, because the photocurrent does not desensitize and therefore generates sustained depolarizations under these conditions. It does raise important questions regarding the effective sensitivity of the system under real-world conditions. Natural viewing presents continuously changing patterned images to the retina that will produce high levels of mean illumination on the retina, much like a white-noise stimulus. If such sustained mean illumination induces a decline in the sensitivity, then the efficacy of the system would be compromised. Since the technology is to be deployed in human patients, it seems important to manage expectations by discussing these potential limitations in more detail. Can this problem be obviated by the design of the external light stimulation device? Is the image projected onto the retina significantly degraded by the proposed solution? How will the approach limit the achievable percept?

Reviewer #2 (Remarks to the Author):

The major claims of the paper are that the newly constructed AAV vectors exhibit improved characteristics compared to the current state of the art, which includes increased sensitivity, and improved infection rates. The results are novel and they will be of interest to others in the regenerative field as a whole and in particular to the field of gene therapy. The work is rather convincing demonstrating by a number of modalities the AAV2.7m8 is an improvement to the previous standard AAV2 and ChR-tdT fusion is better than ChR alone. Overall the data is convincing and the vectors developed will be sought after in the field as a whole. Statistical analyses seem to be conducted appropriately. There are some areas that still have room for improvement. The details in the legends do not allow the reader to understand everything that is being presented.

I do not understand figure 1A. Is the legend reversed? The lighter the signal the less RGCs there are?

Figure legend 1 does not provide enough detail to understand what is happening. What is the insert showing in figure 1B? What are the numbers in figure 1C?

The language is vague, making it difficult to understand what is being said. For example: "These MEA recordings were consistent with a greater efficacy of the AAV2.7m8 - ChR-tdT construct, which was, therefore, systematically used in subsequent experiments." What do they mean by "systematically used"?

Reviewer #3 (Remarks to the Author):

I'm not convinced by the authors' responses. To both of questions, a simple statement was given as answer. No clear analysis or statistical test was provided. I consider this not sufficient as a rebuttal.

Reviewer #3:
Point 1;

Nevertheless, it would be interesting to know whether the injected animals showed any signs of changes vision (e.g. increased light sensitivity or reduced visual capacity due to confusing/altered visual input). As the animals were housed for 2 -6 months, at least obvious effects should have been documented. If no changes were observed this would be a valuable data point, too.

Authors' response:

As suggested by the reviewer, we included information on the animal behavior in Supplementary Materials, Supplementary Materials and methods, in Gene delivery, retina isolation and preservation of the primate retina, page 1, line 16:

"None of the treated animals displayed any signs of photophobia or vision-related changes in behavior."

Reviewer's response: This simply added statement is a bit weak. I would have expected some details here, i.e. how was the lack of vision-related changes assessed?

Reviewer #3:

Point 2

To look into this aspect, close up images of expressing neurons (as in Fig. S3) as well as a comparison of densities of expressing cells as in Figure S1 would be helpful.

Authors' response:

As advised by the reviewer, we examined further the cells at 2 and 6 months (see newly added figure S3). Although we cannot exclude the possibility of a decrease in the number of ChR-tdTomato expressing cells between 2 and 6 months, no change in expression level was detected.

We added the following sentence in the result section, Subhead 2, page 12, line 2:

"We cannot exclude the possibility of a decrease in the number of ChR-tdT-expressing cells between two and six months, but no change in the level of expression was detected (Fig. S3). Furthermore, we observed no difference in the fraction of responsive electrodes (Fig. 2D, 102 ± 58 vs. 73 ± 65 for 2 months and 6 months, respectively, for 5×10^{11} vg/eye), or light sensitivity (Fig.1D and Fig. 2E).

Reviewer's response: How was the lack of change in the level of expression analyzed? Which method and which test was used?

Reviewer #1 (Remarks to the Author):

My comments have been addressed; however, the response to the white-noise receptive field mapping does raise an interesting and pertinent question that was not immediately obvious in my initial reading. The authors note that white-noise analysis was not workable due to “rundown of the efficiency of responses” due to the continuous relatively high mean illumination level. This rundown doesn’t seem to be evident in the photocurrents (Fig. 1G, 4C); suggesting that it likely arises from inactivation of voltage-gated channels in the ganglion cells. There seems to be some cell-to-cell variability (e.g. Fig. 1B, 4F, G), although the data in Fig. 3A suggests that on average spike-accommodation during high continuous illumination is strong. This makes sense, because the photocurrent does not desensitize and therefore generates sustained depolarization under these conditions. It does raise important questions regarding the effective sensitivity of the system under real-world conditions.

The reviewer raises a very pertinent point regarding inactivation of the photocurrent vs voltage-gated channels in the recorded RGCs. We agree with the reviewer that our photocurrent recordings (Fig. 1G, 4C) do not display much inactivation, but a study modeling the ChrimsonR photo-cycle suggested that there is an inactivated state for this microbial opsin (Sabatier et al. 2018, <https://www.biorxiv.org/content/10.1101/417899v1.abstract>). However, it remains unclear whether this state is an intrinsic biophysical property of this opsin, or whether it results from the recycling of the photopigment. Nevertheless, the spiking activity in ChrimsonR-td-tomato-expressing RGCs is modified by long duration of illumination (e.g. 2 s stimuli in Fig. 3). As attested by the fano-factor for the 2 s stimuli (Fig. 3E), prolonged stimulation increases the variability of the spike train generated, decreasing the ability of the cells to encode information reliably. For shorter stimulation durations, the fano factor remains below one, and the variability of spike count is low, demonstrating that the encoding of information is reliable.

Natural viewing presents continuously changing patterned images to the retina that will produce high levels of mean illumination on the retina, much like a white-noise stimulus. If such sustained mean illumination induces a decline in the sensitivity, then the efficacy of the system would be compromised. Since the technology is to be deployed in human patients, it

seems important to manage expectations by discussing these potential limitations in more detail. Can this problem be obviated by the design of the external light stimulation device?

The design of the stimulation device takes this observation into account, by reducing images to their outline. We have introduced the following statement into the paper (Results Subhead 3; page 11, line 14):

“The Fano factor was below 1 for the short duration (1 to 200ms), indicating a lower variability than for the Poisson distribution, but we observed a large increase in spike train variability for stimulations lasting 2 seconds (Fig. 3E). Most of this effect can be attributed to stimulus hysteresis, as retinal sensitivity subsequently recovered. Consistent with this observation, recordings of activity in response to achromatic binary white noise with a 50% pseudorandom selection rate revealed a gradual decline of evoked activity (data not shown). The underlying mechanism of this modulation may involve an inactivated state of the microbial opsin³⁰ or the inactivation of the voltage-gated channels in the ganglion cells. A simple monochrome transformation of natural images would result in a large number of pixels with high values (i.e. light gray) potentially leading to a rapid deactivation of retinal ganglion cells. The goggles used for visual stimulation include an event-based asynchronous camera outlining object contours³¹. It should therefore be possible to overcome the problem of retinal ganglion cell deactivation by reducing the number of active pixels in a projected frame through the limitation of active pixels to object contours. “

Is the image projected onto the retina significantly degraded by the proposed solution?

As noted above, the stimulation device will limit in part the number of active pixels through the use of object contouring, as well as delivery of light in pulse train (see Results Subhead 3, page 12, line 4).

How will the approach limit the achievable percept?

Apart from the fact that it would be limited to object contour, due to the activation pattern used, it is difficult to predict what the actual percept will be in humans. Nonetheless, with the ongoing clinical trial the answer will soon be within our reach.

Reviewer #2 (Remarks to the Author):

The major claims of the paper are that the newly constructed AAV vectors exhibit improved characteristics compared to the current state of the art, which includes increased sensitivity, and improved infection rates. The results are novel and they will be of interest to others in the regenerative field as a whole and in particular to the field of gene therapy. The work is rather convincing demonstrating by a number of modalities the AAV2.7m8 is an improvement to the previous standard AAV2 and ChR-tdT fusion is better than ChR alone. Overall the data is convincing and the vectors developed will be sought after in the field as a whole. Statistical analyses seem to be conducted appropriately. There are some areas that still have room for improvement. The details in the legends do not allow the reader to understand everything that is being presented.

We thank the reviewer for these positive comments.

I do not understand figure 1A. Is the legend reversed? The lighter the signal the less RGCs there are? Figure legend 1 does not provide enough detail to understand what is happening. What is the insert showing in figure 1B? What are the numbers in figure 1C?

The reviewer seems to be referencing figure 2, rather than figure 1. Nevertheless, both figures have been modified and their legends extended, to make them easier to read. Figure 2 A did indeed contain an error, with a reversal of the color bar. This has now been corrected. The inset in Figure 2B shows the profiles of RGC density for individual retina for the three vector doses. The numbers in Figure 1C indicate the light intensity used. We have also modified the legends of the other figures. Figures 5, 6 and 7, in particular, have been developed further, to improve the explanations provided and to clarify the data presented. All modifications are shown in TrackChanges mode in the submitted document.

We have also modified these figures as follows:

- Fig. 1-A: the colors of the markers have been changed and a dashed outline of the retina has been added.
- Fig. 1-B: the number of responsive electrodes at the top has been replaced by the number of responsive electrodes/number of active electrodes.
- Fig. 1-F: a scale bar has been added.

- Fig. 1-H: the color of the 2nd axis is represented as a grayscale.
- Fig. 2-A: the reversed color bar has been corrected.
- Fig. 2-B inset: a X and Y axis label have been added as well as vector dose on the graph.
- Fig. 2-C: The label of the x-axis has been changed from “seconds” to “Time (s)”. We added “photons.cm⁻².s⁻¹” on top of the stimuli to indicate that numbers are irradiance.
- Fig. 4-A: We have added an outline of the retina explants as a dashed line, adjusted the rectangle accordingly, and displayed a scale bar.
- Fig. 1, 2, 4: we have changed the illuminance units to match a single nomenclature (photons.cm⁻².s⁻¹) and the exponent is expressed correctly (e.g. 5x10¹¹).
- Fig. 1, 2, 3, 6, 7: all units are converted to SI units (mostly “sec” replaced by “s” or “msec” replaced by “ms”).

We have also made minor esthetic modifications.

The language is vague, making it difficult to understand what is being said. For example: "These MEA recordings were consistent with a greater efficacy of the AAV2.7m8 - ChR-tdT construct, which was, therefore, systematically used in subsequent experiments." What do they mean by "systematically used"?

As mentioned in our reply to the Editor, we have made a number of changes and native English-speaking professional scientific editor has reviewed our manuscript. The sentence (Results Subhead 1, page 8, line 6): “These *MEA recordings were consistent with a greater efficacy of the AAV2.7m8 - ChR-tdT construct, which was, therefore, systematically used in subsequent experiments*” was modified to facilitate understanding: “These *MEA recordings were consistent with a greater efficacy of the AAV2.7m8-ChR-tdT construct; this construct was therefore used in all subsequent experiments.*”

Reviewer #3 (Remarks to the Author):

I'm not convinced by the authors' responses. To both of questions, a simple statement was given as answer. No clear analysis or statistical test was provided. I consider this not sufficient as a rebuttal.

In the new version of the manuscript, we have attempted to address the points raised by the reviewer. We also want to report that we added a supplementary figure (Fig. S3) when we resubmitted our manuscript to address Reviewer #3's comments.

Reviewer #3:

Point 1;

Nevertheless, it would be interesting to know whether the injected animals showed any signs of changes vision (e.g. increased light sensitivity or reduced visual capacity due to confusing/altered visual input). As the animals were housed for 2 -6 months, at least obvious effects should have been documented. If no changes were observed this would be a valuable data point, too.

Authors' response:

As suggested by the reviewer, we included information on the animal behavior in Supplementary Materials, Supplementary Materials and methods, in Gene delivery, retina isolation and preservation of the primate retina, page 1, line 16:

"None of the treated animals displayed any signs of photophobia or vision-related changes in behavior."

Reviewer' response: This simply added statement is a bit weak. I would have expected some details here, i.e. how was the lack of vision-related changes accessed?

Author's response:

We agree with the reviewer that any behavioral change in the animals following treatment could have been a sign of altered visual input. However, housing conditions cannot trigger optogenetic activation, as attested by the newly added quantification of light intensity in housing unit. We added this information in the Material and Methods (Section Animals, page 25, line 17):

“Animals are housed in a room lit with fluorescent tubes which is filtered through diffuser panels above each housing unit to deliver a cool white light (4000K), during a 12-hour photoperiod. The maximum amount of light that could be measured, corresponding to an animal looking directly at the light in the upper part of its enclosure, is 585 $\mu\text{W}/\text{cm}^2$. This maximal possible exposure does not reach the light intensity threshold for activating ChrimsonR, measured here at 2.34×10^{15} photons. $\text{cm}^{-2}.\text{s}^{-1}$ (784 $\mu\text{W}/\text{cm}^2$, Fig. 1D & 2E). Furthermore, the light to which the animals are exposed in these housing conditions is a broad white light with a large spectrum that would be unable to activate ChrimsonR in an optimal manner, unlike the pure 595 nm orange light used in the study. We therefore expected to see no behavioral changes due to ChrimsonR activation in these normal housing conditions. We nevertheless continually monitored the behavior of the animals, by checking that their feeding habits were normal, and evaluating general stress and behavior during enrichment periods. None of the treated animals showed any sign of stress due to photosensitivity. No noticeable changes of behavior (during feeding and enrichment) were observed after the surgery.”

We moved the following sentence from the supplementary materials and methods (section: Gene delivery, retina isolation and preservation of the primate retina) to Results Subhead 1 (Results Subhead 1, page 6, line 5):

“None of the treated animals displayed any signs of photophobia or vision-related changes in behavior under normal lighting in the animal house”.

Reviewer #3 Point 2

To look into this aspect, close up images of expressing neurons (as in Fig. S3) as well as a comparison of densities of expressing cells as in Figure S1 would be helpful.

Authors' response:

As advised by the reviewer, we examined further the cells at 2 and 6 months (see newly added figure S3). Although we cannot exclude the possibility of a decrease in the number of ChR-tdTomato expressing cells between 2 and 6 months, no change in expression level was detected.

We added the following sentence in the result section, Subhead 2, page 12, line 2:

“We cannot exclude the possibility of a decrease in the number of ChR-tdT-expressing cells between two and six months, but no change in the level of expression was detected (Fig. S3). Furthermore, we observed no difference in the fraction of responsive electrodes (Fig. 2D, 102 ± 58 vs. 73 ± 65 for 2 months and 6 months, respectively, for 5×10^{11} vg/eye), or light sensitivity (Fig.1D and Fig. 2E).

Reviewer’ response: How was the lack of change in the level of expression analyzed? Which method and which test was used?

Author’s response:

In response to the reviewer’s original comment we added a new supplementary figure with epifluorescence, confocal and 2-photon images of the primate’s retina at 2 and 6 month post-injection (Figure S3). Within this figure, we can clearly see that the transfection pattern is similar two and six months post-injection, in terms of the spread (epifluorescence images on the left of the figure), cellular localization of fluorescence (labeled membrane and cytosol) and healthy cell morphology (no cellular debris, swollen neurites or td-tomato-positive nuclei). Unfortunately, we were unable to acquire large stitches of high-resolution images for the 2-month time point, and we cannot, therefore, quantify the number of expressing cells precisely. The comparison of constructs at the 2-month time point was based on the activity generated by ChR-tdT (Fig. 1C-D) rather than fluorescence expression, as two of the constructs considered produced no fluorescent protein. This is the reason we compared 2-months and 6-months time points using the proportion of active electrode responding to light flashed in presence of blockers (Fig. 2D).

Our assertion (Results Subhead 2, page 9, line 18):

“We cannot exclude the possibility of a decrease in the number of ChR-tdT-expressing cells between two and six months, but no change in the level of expression was detected (Fig. S3). Furthermore, we observed no difference in the fraction of responsive electrodes (Fig. 2D, 102 ± 58 vs. 73 ± 65 for 2 months and 6 months, respectively, for 5×10^{11} vg/eye), or light sensitivity (Fig.1D and Fig. 2E).”

Has been modified as follows:

“We cannot exclude the possibility of a decrease in the number of ChR-tdT-expressing cells between two and six months, as we were unable to obtain cell counts for both ime points.

However we observed no major differences in the expression profile on the fovea, and no changes in the subcellular pattern of expression (Fig. S3). Furthermore, we observed no difference in the fraction of responsive electrodes (Fig. 2D, 102 ± 58 vs. 73 ± 65 for 2 months and 6 months, respectively, for 5×10^{11} vg/eye), or light sensitivity (Fig.1D and Fig. 2E)."

REVIEWERS' COMMENTS:

Reviewer #1 (Remarks to the Author):

The authors have satisfactorily addressed my comments.